# Education shapes the link between EEG aperiodic components and cognitive aging

Sara Lago[1,2*], Sara Zago[1], Sonia Montemurro[3], Rocco Salvatore Calabrò[4],
Maria Grazia Maggio[4], Serena Dattola[4], Ilaria Casetta[1], Giorgio Arcara[1,5*]

1 IRCCS San Camillo Hospital, Venice, Italy, 2 University of Padova, Padova, Italy, 3 Department of Philosophy, Sociology, Education and Applied Psychology (FISPPA), University of Padova, Padova, Italy, 4 IRCCS Centro Neurolesi "Bonino Pulejo", Messina, Italy, 5 Department of General Psychology, University of Padova, Padova, Italy

* sara.lago@hsancamillo.it (S.L.); giorgio.arcara@unipd.it (G.A.).

## Abstract

Healthy aging brings widespread shifts in aperiodic (non-oscillatory) electroencephalographic (EEG) components, which may underlie physiological changes in cognitive performance. Education, a known protective factor against age-related decline in cognitive performance, has been largely overlooked in studies linking aperiodic EEG components to cognition. This study addresses this gap, hypothesizing that education moderates the interplay between age, aperiodic components, and cognitive performance, as measured by Mini-Mental State Examination (MMSE) scores. We reanalyzed an open-source EEG dataset of 714 healthy individuals aged 18–91 years using Generalized Additive Mixed Models. Aperiodic components (exponent and offset) declined with age, but higher education levels mitigated these declines. Notably, the aperiodic components interacted with age and education in predicting MMSE performance in a widespread way across the cortex. Among older adults, the relationship between the aperiodic components and cognitive performance diverged by education: those with lower education showed worse cognitive outcomes with lower exponents and offsets, whereas higher-educated individuals after 60 years showed a reverse pattern, with lower exponents and offsets predicting better MMSE performance. Our findings suggest that the link between aperiodic components and cognitive aging is not straightforward but depends on moderating factors such as education. These results underscore the importance of accounting for individual differences, like educational background, when exploring age-related changes in EEG aperiodic components and cognition.

## 1. Introduction

Recent research on the EEG signal has shifted from the traditional focus on oscillatory activity in specific EEG frequency bands to a growing interest in the aperiodic (non-oscillatory) component. In the electrophysiological signal, oscillatory activity is

**Data availability statement:** All relevant data can be found at https://github.com/euroladbrainlat/Brain-health-in-diverse-setting.

**Funding:** The author(s) received no specific funding for this work.

**Competing interests:** The authors declare no competing interests.

associated with peaks that can be observed in the Power Spectrum Density (PSD). However, the PSD is always characterized by a general (and broadband) component, that has been named recently "aperiodic component", characterized by a 1/f-like distribution. Separating the aperiodic and the periodic component is now a relevant perspective with important implications in the interpretation of results. Historically, neural activity has been measured by averaging power within predefined frequency bands derived from the power spectrum, thus confounding the periodic with the aperiodic component [1]. The aperiodic component of the EEG power spectrum is typically described by two parameters: the exponent, capturing the steepness of the spectral decay, and the offset, reflecting the broadband vertical shift of the PSD [1, 2]. These parameters are sometimes interpreted as having different and separate neurophysiological underpinnings, e.g., excitation-to-inhibition (E:I) balance or neural spiking rate respectively [3–6], but they are mathematically related: when a rotational pivot is present around a non-zero frequency of the PSD, changes in the exponent necessarily induce changes in the estimated offset, leading to strong empirical correlations between the two [1, 4, 6–8]. Thus, exponent-offset correlations are not competing with interpretations of these components as having separate neural generators, but rather expected properties of the model. At the same time, the offset can vary independently of the exponent, for example in cases where overall broadband power increases or decreases without a change in spectral slope. In such cases, the offset may reflect distinct physiological processes (i.e., neural spiking rates) [5]. However, this independence must be demonstrated empirically, and offset should not be interpreted separately from the exponent unless these conditions are met. In addition, the relation with a specific neurophysiological underpinning is questioned and several changes in neurophysiological properties (e.g., presence of bursts, or noise in the signal) can affect the aperiodic component of the spectrum [1, 9, 10]; in this vein, recent biophysical and pharmacological work indicates that the spectral exponent is shaped by multiple mechanisms—including synaptic kinetics, nonlinear membrane dynamics, and aperiodic network activity—and therefore may not map one-to-one onto the E:I ratio (e.g., [11, 12]). In this view, the exponent is better treated as a sensitive but non-specific marker of circuit state, to be interpreted alongside offset and oscillatory features rather than as a direct assay of E:I. Relatedly, recent modelling cautions that choices about spectral detrending—i.e., removing the broadband 1/f background from the PSD before quantifying narrowband peaks—(e.g., subtractive vs divisive) should be physiologically justified, particularly under pharmacological manipulations, because inappropriate detrending can bias oscillatory power estimates [11].

Recent studies have demonstrated the functional significance of aperiodic components, associating variations in its parameters with task performance [3, 13, 14], age-related cognitive decline [15–18], and brain disorders [19, 20]. Consequently, the aperiodic component is increasingly recognized as a critical biomarker for understanding neural processes and cognitive function [19, 21], but at the same time, evidence from state manipulations (e.g., sedation vs excitatory drugs administration [12]; different sleep stages [3]) suggests that behavioral state can modulate the exponent independently of a simple E:I shift, reinforcing the need for cautious interpretation.

Aging is associated with widespread changes in aperiodic EEG components that reflect non-pathological alterations in neuronal networks [15, 17]. Both the exponent and offset of the aperiodic component tend to decline with age, indicating shifts in the excitation-to-inhibition (E:I) balance and possibly reductions in overall neural firing rates [4, 15, 17, 22], even though recent literature reports contradicting results [23]. Given these considerations, age-related slope "flattening" is compatible with broader biophysical changes (e.g., synaptic time scales, leak currents, nonlinearities) and should not be ascribed exclusively to E:I alterations [11].

Age-related changes in aperiodic components contribute to cognitive performance throughout the lifespan. Specifically, the reduction in the exponent (often referred to as the flattening of the aperiodic slope) has been linked to age-related declines in cognitive functions, such as working memory [17]. Subsequent studies have replicated these findings and linked flatter slopes in older adults to poorer performance on spatial attention, short-term memory, processing speed, and executive function tasks [14, 16, 24, 25]. Additionally, beyond the exponent, the offset has been found to be negatively correlated with reaction time, perceptual sensitivity, processing speed, and selective attention performance [13, 25–27]. These findings support the neural noise hypothesis, which posits that aging-related reductions in the exponent reflect increased desynchronized background neural activity, or neural noise, driven by shifts in the E:I ratio. This increased noise disrupts neural communication fidelity, leading to a flatter power spectrum and reduced efficiency in cognitive processing [17]. Consequently, decreases in the exponent during healthy aging may indicate greater neural excitability and diminished E:I balance, aligning with broader patterns of age-related neurophysiological change [28]. However, some findings contradict these results. Euler et al. [26] reported no association between the aperiodic exponent and specific cognitive constructs such as working memory, perceptual reasoning, processing speed, and verbal comprehension in participants aged 18–52. However, they observed a relationship between the exponent and an aggregated measure of cognitive ability. Similarly, Cesnaite et al. [29] found no link between aperiodic components and cognitive performance in an older cohort (aged 60–80), while Smith et al. [30] reported that higher exponents were associated with better general cognitive function but were unrelated to age (50–80). Such heterogeneity may reflect differences in task demands, age ranges, education, and unmeasured state-related factors that can influence aperiodic estimates. Consistent with these diverse findings and with recent critiques, we adopt a conservative stance and refrain from assigning a unique mechanistic meaning to exponent shifts; instead, we evaluate exponent and offset jointly and in the context of behavioral state and oscillatory markers [11, 12].

That said, associations between cognition and aperiodic components are not consistent across all ages and tasks, suggesting that the relationship between aperiodic components and cognition is task- and age-dependent [29, 31] and highlighting the possible influence of unexamined moderating variables such as education. Education is a well-established predictor of cognitive performance, with higher educational attainment typically associated with better cognitive outcomes [32, 33]. While some studies found no moderating effect of education on the relationship between aperiodic components and cognition [29], preliminary findings suggest otherwise. Montemurro et al. [18] reported that in older adults, the relationship between the aperiodic exponent and cognitive performance varied by education: higher exponents were predictive of worse processing speed and working memory in highly educated individuals, but better performance in those with lower education. The authors suggest that the relationship between exponent, neural noise, and cognitive performance may not be straightforward, and highlight the importance of investigating possible mediators, such as education, within this complex relationship. However, this study categorized participants into discrete groups (e.g., younger vs. older adults, higher vs. lower education), treating inherently continuous variables as categorical. This approach reduces data variability and may obscure nuanced relationships between education, aging, and aperiodic components.

To address such existing limitations and further explore the potential moderating role of education in the relationship between aperiodic components and cognition in aging, we reanalyzed data from a publicly available dataset [34] (accessible at https://github.com/euroladbrainlat/Brain-health-in-diverse-setting). This dataset includes measures of

aperiodic EEG components (exponent and offset), age, education, and Mini-Mental State Examination (MMSE) scores [35] as continuous variables for 714 participants aged 18–91 years, thus covering a wider age range than previous studies [26, 29, 30].

The present study aimed to achieve two objectives: [1] to examine the relationships between EEG aperiodic components, age, and education, with the goal of confirming established age-related changes and investigating less-studied effects of education on aperiodic components, and [2] to investigate whether demographic variables (age and education) and EEG aperiodic components could predict cognitive performance measured with the MMSE. We hypothesized that education may serve as a protective factor against age-related decline in aperiodic exponent and offset. Furthermore, we posited that this protective effect might extend to cognitive abilities, such that participants with higher educational attainment would exhibit higher aperiodic exponents and offsets, alongside better MMSE performance, compared to their less-educated counterparts. In line with current debates, we quantified the aperiodic exponent and offset separately, but we interpret them jointly, acknowledging their strong interdependence due to the mathematical property of power spectrum rotation: when a change in the exponent induces a pivot around a non-zero frequency, the estimated offset shifts as well; [7, 36] thus, the correlation between exponent and offset is not a competing interpretation to other explanations in the literature (exponents and offsets reflecting different neurophysiological processes), but rather an inherent property of the spectral parametrization model. We therefore refrain from assigning parameter-specific physiological mechanisms and instead frame our results in terms of general aperiodic dynamics, consistent with multiple plausible biophysical interpretations [11, 12].

## 2. Materials and methods

### 2.1. Participants

Demographic and cognitive data for the sample are summarized in Table 1. From the original larger dataset, a subsample of 714 participants for which MMSE and aperiodic EEG data were available were included in our analyses. Participants had no history of psychiatric or neurological disorders, alcohol or drug abuse, significant sensory impairments, or functional cognitive complaints. Participants' age ranged from 18 to 91 years, while their education was encoded as the number of completed years of formal education and ranged from 0 to 26 (see Fig 1 for the distributions of these variables). Both age and education were encoded as numerical, continuous variables (this being especially important for education, as a given number of years of education may map to different qualifications internationally). Education levels were uniformly distributed by age (see Supporting Information). The MMSE [35] was used to assess participants' general cognitive state. MMSE is a widely used screening tool for cognitive impairment that evaluates various cognitive domains, including orientation to time and place, short-term memory recall, working memory, language abilities, visuoconstructional skills, and basic motor commands. The MMSE consists of 11 items, with a score range from 0 to 30 (generally, a score of 24 or higher indicates normal cognitive functioning [37]).

Additional details about participants are provided in Hernandez et al. [34]. Since the study is based on an anonymized dataset, no application to the Ethics Committee was made. In the original article [34] an application to the internal Ethics Committee was obtained prior to data collection, in compliance with the Helsinki declaration.

**Table 1. Demographics and cognitive information for the participants' sample.**

| Sex | n | Age mean (SD) | Education mean (SD) | MMSE mean (SD) |
|---|---|---|---|---|
| F | 413 | 58.390 (15.209) | 13.169 (4.468) | 28.630 (1.862) |
| M | 301 | 52.472 (20.320) | 13.500 (3.753) | 28.804 (1.786) |

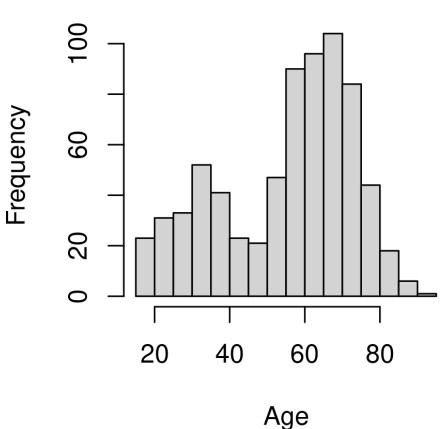
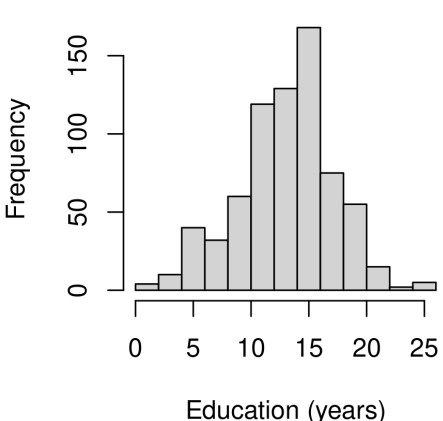

**Fig 1. Histogram distributions for age and education.**

## 2.2. EEG acquisition and processing

Details of EEG acquisition, preprocessing, source estimation, and aperiodic component extraction are described in Hernandez et al. [34]. Briefly, participants were seated in comfortable chairs within dimly lit, electromagnetically quiet rooms, and were advised to stay still and alert while resting-state EEG was recorded (both in eyes-closed and eyes-open conditions). The original paper reported no information regarding the timing or sequence of data collection (e.g., whether EEG and cognitive scores were obtained in close temporal proximity). Since the original study was a multicentric one and the recording duration could vary across centers, to uniform recording duration only the first 5 minutes were selected. After EEG recording and preprocessing, EEG data was filtered between 0.5 and 40 Hz. Source reconstruction was performed using the standardized low resolution electromagnetic tomography (sLORETA) algorithm [38], projecting the EEG signal on the Montreal Neurological Institute (MNI) template. The source-reconstructed signal was segmented into 82 regions using the Automated Anatomical Labeling (AAL) atlas [39].

The aperiodic exponent and offset were derived from the source-reconstructed EEG signal using the FOOOF algorithm, which models the aperiodic component with a Lorentzian function:

$$L = b - log(k + Fx)$$

where $b$ represents the broadband offset, $\chi$ is the exponent, and $k$ is the 'knee' parameter, which accounts for the bend in the aperiodic component (not considered in the present analyses). F denotes the vector of input frequencies [1]. The algorithm covered the frequency band from 0.5 to 40 Hz, with peak width limits from 1 to 6 Hz, a maximum number of peaks of 6, a minimum peak height of 0.2, and peak threshold of 2.0 [34].

To generate comparable ROIs, and therefore facilitating data interpretation and enhancing the identification of patterns in the results, Hernandez et al. aggregated the AAL regions into ten composite regions of interest (ROIs) via mean averaging. Their approach involved two main criteria: [1] grouping together brain regions associated with a specific cortical gyrus (e.g., superior, middle, and inferior orbital gyri) into a single ROI, and [2] assembling neighboring regions with established functional coupling, such as the Rolandic operculum and insula, with the goal of concentrating on ROIs that include both structurally and functionally related regions, rather than analyzing each small region in isolation. The composite ROIs and their corresponding anatomical regions are presented in Table 2. The dataset did not include information on the aperiodic components in the frontal brain areas nor about the quality of the aperiodic fit such as the $R^2$.

**Table 2. Corresponding brain regions for each composite ROI (adapted from Hernandez et al. [34]).**

| Composite ROI | Regions comprising the ROI |
|---|---|
| left and right cingulate cortex (CING) | Anterior, middle, and posterior cingulate |
| left and right Hippocampal Regions and Amygdala (HPC) | Hippocampus, para-hippocampal area, amygdala |
| left and right parietal lobe (parietal) | Superior and inferior parietal gyri, angular gyrus, paracentral lobule |
| left and right occipital lobe (occipital) | Calcarine fissure and surrounding cortex, cuneus, superior, middle, and inferior occipital gyri, lingual gyrus |

## 2.3 Statistical analyses

Statistical analyses were conducted using R (version 4.4.1; 14.06.2024; [40]). Generalized Additive Mixed Models (GAMMs) were implemented using the mgcv package (version 1.9.1; [41]). GAMMs are nonlinear mixed-effects regression methods [42, 43] and were chosen because they allow to model nonlinear interactions between variables. Moreover, they can also accommodate continuous variables [44], a crucial feature when investigating cognitive and brain aging such in the present case: Thuwal et al. [24] pointed out that changes in exponent are associated in a non-linear way with age, and aging is indeed a continuous process, that accumulates effects gradually, and can assume different individual trajectories [45]. Therefore, these features need to be considered in the statistical model.

Separate models were computed for each region of interest (ROI) and each aperiodic EEG component (exponent and offset). To account for individual differences, all models included a random factor smooth for subjects, enabling nonlinear adjustments to regression shapes [43]. The default Gaussian distributional family was used in all models.

For both research aims, a model comparison strategy was employed to identify the most appropriate statistical model for each ROI. Separate simple and complex models were constructed for each aim and ROI.

For Aim 1 (exploring relationships between EEG components, age, and education) the following models were constructed:

Simple: EEG component ~ sex + s(Age) + s(Education)

Complex: EEG component ~ sex + s(Age) + s(Education) + ti(Age, Education)

The simple model evaluated the main effects of age and education, while the complex model additionally considered their interaction.

On the other hand, for Aim 2 (exploring how changes in MMSE scores relate to EEG components, age, and education) the following models were constructed:

Simple: transformed MMSE scores ~ sex + s(EEG component) + s(Age) + s(Education)

Complex: transformed MMSE scores ~ sex + s(EEG component) + s(Age) + s(Education) + ti(EEG component, Age) + ti(EEG component, Education) + ti(Age, Education) + ti(EEG component, Age, Education)

In Aim 2, the simple model did not include any EEG components and was therefore independent of specific ROIs. This model captured known patterns of cognitive change across increasing age and education levels and served as a "baseline" against which all other complex models were compared.

A key strength of GAMMs is their capacity to model random effects capturing variability across participants and experimental items. In the present study, random effects were not included because the dataset was not nested (each participant contributed only a single observation), and preliminary analyses indicated that incorporating participant-level random effects introduced non-normality in the residuals.

In all models, sex was coded as a factor and contrasts were set to 0 = F (female) and 1 = M (male). In the GAMM syntax, the term 's()' is used to set in the model the main effect of a variable, while 'ti()' is used for setting an interaction effect of two or more variables. The number of basis functions (the k parameter) was set to the default values: 9 for main effects, 16 for two-way interactions, and 64 for three-way interactions. Given the degree of the models' complexity, these values provided sufficient model flexibility given the available data while preventing overfitting (see the Deviance Explained columns of Tables 3–6). To check that these k settings were adequate, we then re-fit the models doubling the value of k and inspected the models' residuals structure using the gam.check function to ascertain if there was any pattern in the residuals that could potentially be explained by increasing k (as advised in https://stat.ethz.ch/R-manual/R-devel/library/mgcv/html/choose.k.html and in Wood [42]). In no case increasing k provided benefits in removing any structure in the residuals, so we went on to use the default k settings.

It is worth noticing that MMSE scores exhibit a strong left-skew due to the high prevalence of scores near the upper limit [30] among healthy participants, particularly younger individuals. Including such a skewed variable in the models caused significant deviations from normality in the models' residuals, which compromised the reliability of inferences derived from the results. To address this issue and bring the residuals closer to normality, MMSE scores were reversed and log-transformed prior to their inclusion in the model.

Model comparisons were conducted using the Akaike Information Criterion (AIC) [46], and the difference between the AIC of the complex and the simple model for each ROI was computed. In some cases (especially for Aim 1), the simple and complex versions of a model resulted in very similar AICs, i.e., in very small differences. To adopt a conservative approach and to avoid commenting on minimal differences that might indicate a negligible superiority of the complex model on the simple one, we chose to only consider differences of more than −2 to indicate a better fit for the more complex model. In light of this, only the results from the best-fitting complex models (those with an AIC difference greater than −2) are reported here, while results for simpler models are available in the Supporting Information. For completeness, Maximum Likelihood (ML) scores, model comparisons based on ML, and their corresponding p-values are also reported alongside AIC values for each model. However, model selection was based solely on AIC differences.

**2.3.1 Significance Testing for predicted effects.** Significance testing followed the approach of Sóskuthy [47]. This approach is mainly based on the visual interpretation of each model's main effects or interaction surfaces and the relative surface differences, while the main effects' and interactions' (also called smoothers) p-values are not considered. After identifying the best-fitting model based on AIC, for simple models significant main effects were assessed by plotting smoothers (available in the Supporting Information) with a customized version of the plot_smooth function. For Aim1, these plots illustrated aperiodic components depending on age or education as continuous variables (terms s(Age) and s(Education) in the models); for Aim 2, they illustrated MMSE scores depending on aperiodic components (term s(EEG component)), age or education. In both cases, they resulted in 2D graphs.

For complex models instead, the interactions' plots resulted in 3D graphs (surface plots). For Aim 1, they depicted aperiodic components depending on the two-way association between age and education (term ti(Age, Education), Figs 1 and 3), while for Aim 2 they depicted the MMSE scores depending on three-way association between aperiodic components, age and education (term ti(EEG component, Age, Education), Figs 5a and 6). Surface plots were produced using a customized version of vis.gam. In the surface plots, lighter yellow and green shades correspond to higher values of the response variable (exponent, offset or MMSE score), while darker blue shades correspond to lower values of the response variable. White-shaded areas indicate regions where the confidence intervals (95%) around the predicted surface included zero, i.e., where the interaction was not significant.

Since the interpretation of such 3D graphs can be confusing, it is possible to break them down by selecting different values of education (for Aim 1 plots) or aperiodic components (for Aim 2 plots). This allows us to plot 2D graphs "zooming in" on the dependent variable (aperiodic component or MMSE score, as per the different aims) in relation with selected levels of an independent variable: we can examine how exponent and offset change according to different education

levels with increasing age (Aim 1, Figs 2a-b, 4a-d) and how MMSE scores change according to different education and exponent levels with increasing age (Aim 2, Figs 7a-d). These plots were obtained with the plot_smooth function.

From these 2D representations we can compute significant differences in the dependent variable according to the different selected levels of the independent variable. In our case, for Aim 1 we can examine whether there is a significant effect of education on exponent and offset, and at which age ranges this difference is significant (Figs 2c-d, 4e-h; 4m-p); in the same vein, for Aim 2 we can test whether and at which age range the interaction between education and exponent has a significant effect on MMSE scores (Figs 7e-h). In these difference plots, age ranges of significant differences are highlighted with a red line on the x axis. These plots were obtained using the plot_diff function. [40]. The R functions for plotting and significance testing are available in the package itsadug (version 2.4.1; [48]).

## 3. Results

### 3.1. Aim 1: relationships between aperiodic components, age and education

**3.1.1. Exponent.** When analyzing the exponent, a complex model incorporating the interaction term was found to be preferable for the left cingulate, bilateral HPC, left parietal and left temporal regions. In the remaining regions, the differences in AIC between models were negligible (Table 3).

Age and education modulated the exponent differently in the left cingulate, bilateral HPC, left parietal and left temporal regions, as illustrated in Fig 2, where higher exponents are represented by lighter yellow and green shades, while lower exponents are represented by darker blue shades. White-shaded areas instead represent regions of the plot where the 95% confidence intervals around the predicted surface included zero, i.e., the interaction was not significant. Younger individuals generally exhibited higher exponents, which declined with age (in Fig 2a-e, the lighter shades in correspondence to young age become darker with increasing age). Importantly, education appeared to mitigate these age-related decreases in different ways across brain areas: in the left cingulate and left parietal (in the upper right corner of Fig 2a and 2d), darker blue in correspondence to older age and lower education becomes lighter with increasing education; in the left HPC and left temporal (Fig 2b and 2e), blue becomes green/yellow in correspondence with education levels around 10 and 20–25 years; in the right HPC (Fig 2c), we observe a similar modulating effect of 20 years of education lasting across the lifespan.

**Table 3. ML and AIC scores (including differences between complex and simple models), along with effective degrees of freedom (Edf) and p-values assessing the significance of differences between the simple and complex exponent models. Percentages of explained deviance are also reported. Model selection is only based on AIC differences greater than −2 (marked with \*).**

| ROI | Explained Deviance | | ML scores | | | AIC | | | Edf | | | |
| | complex | simple | complex | simple | ML score difference (c-s) | complex | simple | AIC difference (c-s) | complex | simple | df | p value |
|---|---|---|---|---|---|---|---|---|---|---|---|---|
| left cingulate | 5.797 | 5.493 | 1048.932 | 1052.034 | 3.102 | 2094.161 | 2096.46 | −2.299* | 9 | 6 | 3 | 0.102 |
| right cingulate | 7.361 | 6.75 | 1049.18 | 1051.653 | 2.473 | 2093.384 | 2094.744 | −1.36 | 9 | 6 | 3 | 0.176 |
| left HPC | 17.098 | 15.076 | 994.828 | 999.06 | 4.232 | 1974.201 | 1983.081 | −8.88* | 9 | 6 | 3 | 0.37 |
| right HPC | 15.426 | 14.455 | 973.494 | 976.452 | 2.958 | 1933.123 | 1936.385 | −3.262* | 9 | 6 | 3 | 0.116 |
| left occipital | 10.99 | 10.071 | 1022.126 | 1022.819 | 0.692 | 2035.979 | 2035.81 | 0.169 | 9 | 6 | 3 | 0.709 |
| right occipital | 10.389 | 10.27 | 1023.946 | 1025.002 | 1.056 | 2042.909 | 2041.873 | 1.036 | 9 | 6 | 3 | 0.549 |
| left parietal | 12.945 | 12.305 | 1014.771 | 1016.758 | 1.988 | 2018.769 | 2021.041 | −2.272* | 9 | 6 | 3 | 0.264 |
| right parietal | 8.061 | 7.962 | 1043.206 | 1045.775 | 2.57 | 2083.097 | 2084.347 | −1.25 | 9 | 6 | 3 | 0.162 |
| left temporal | 22.11 | 19.908 | 969.633 | 972.599 | 2.965 | 1915.394 | 1925.452 | −10.058* | 9 | 6 | 3 | 0.115 |
| right temporal | 18.183 | 17.923 | 951.725 | 952.729 | 1.004 | 1887.262 | 1887.256 | 0.006 | 9 | 6 | 3 | 0.571 |

# Exponent complex models: Age*Education interaction

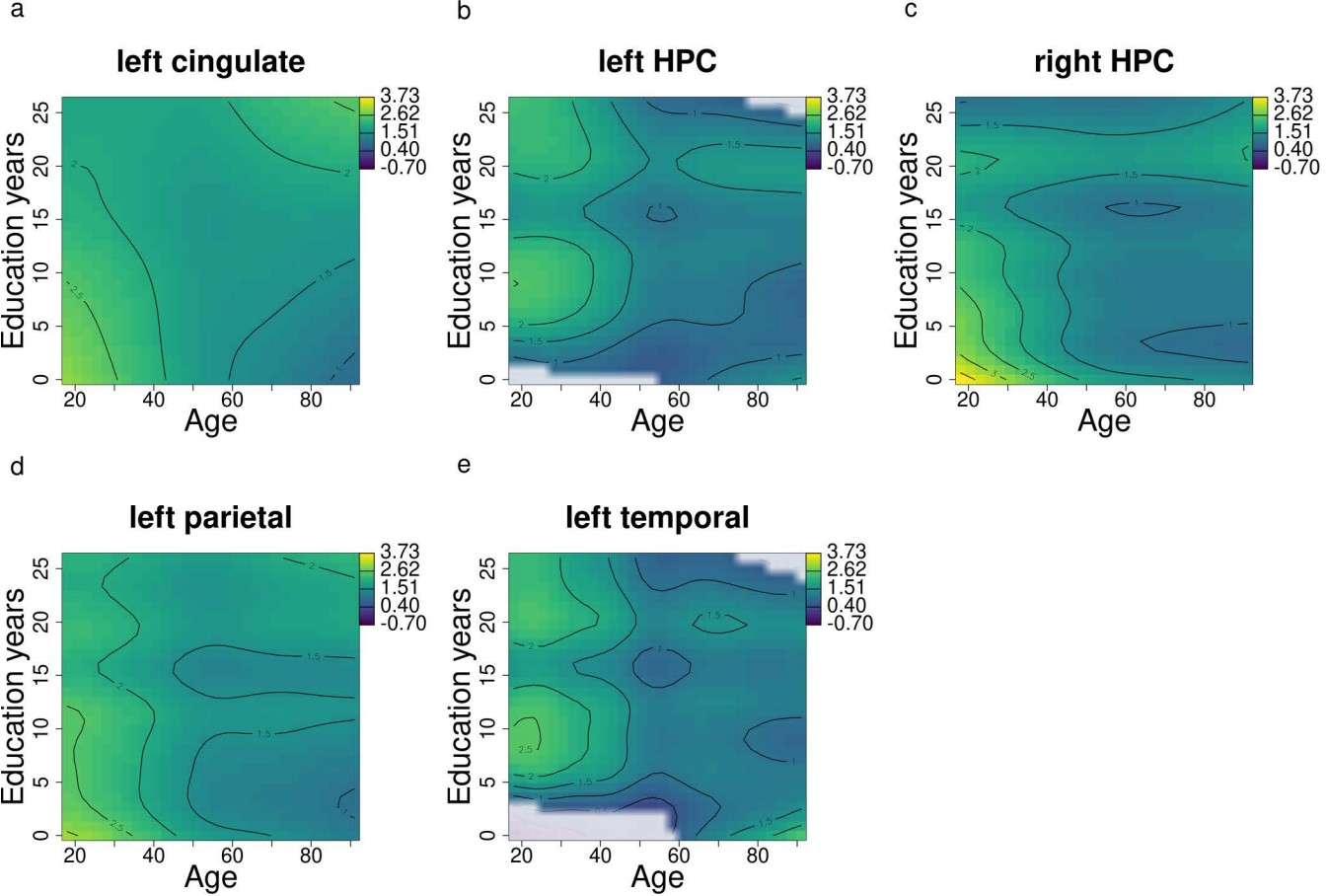

**Fig 2. Exponent changes over age and education (results of the complex models' interactions) for the left cingulate, bilateral HPC, left parietal and left temporal areas.** Darker blue shades indicate lower exponents, while lighter green and yellow shades indicate higher exponents. White-shaded areas indicate regions where the confidence intervals (95%) around the predicted surface included zero, i.e., the interaction was not significant.

Comparisons between individuals with high and low education levels (Fig 3) revealed that significant differences in exponents emerged in all the above-mentioned regions between 40 and 60 and lasted until 91 years of age (Fig 3d–f, j), except for the left temporal region, where significant differences in exponents span 40–70 years approximately. This suggests that in the left cingulate, bilateral HPC, left parietal and left temporal areas, education significantly modulates the relationship between age and exponent, with higher education levels associated with less pronounced declines in exponent values over time.

In summary, we found that older adults exhibit lower exponents (flatter slopes) in the left cingulate, bilateral HPC, left parietal and left temporal areas. Furthermore, middle-aged and older adults with higher education demonstrated significantly higher exponents compared to their peers with lower education levels.

**3.1.2. Offset.** When analyzing the offset, a complex model incorporating the interaction between age and education was found to be preferable over a simpler model in the left cingulate and bilateral HPC. As for the exponent, in the remaining regions either the differences in AIC between models were negligible (Table 4).

# Exponent by education differences

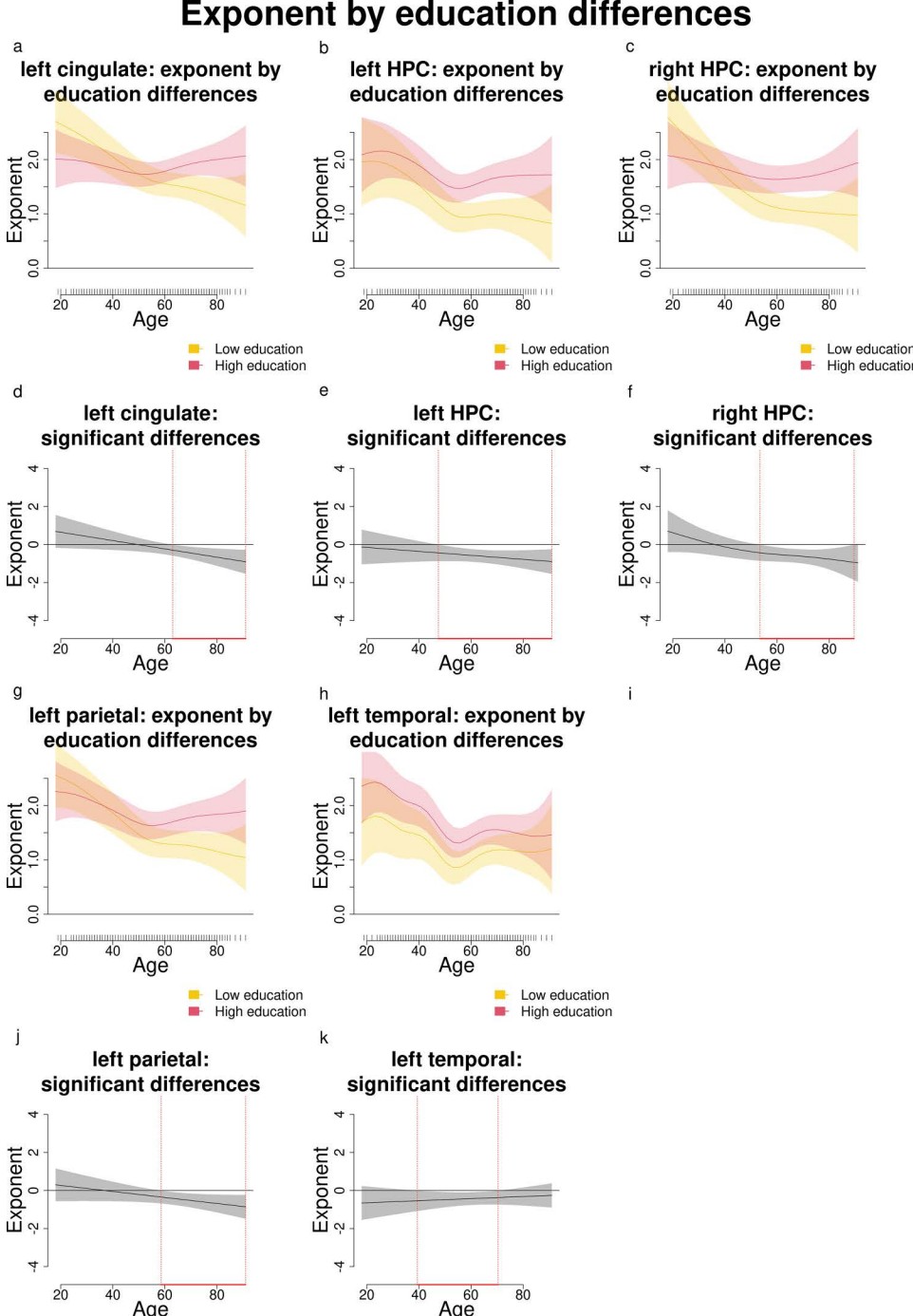

**Fig 3. Comparisons and significant differences between exponent levels of participants with high and low education, in relation with age for the left cingulate, bilateral HPC, left parietal and left temporal areas.** Panels a, b, c, g and h show how age shapes the exponent for different levels of education (high and low). Panels d, e, f, j and k show significant differences in exponent levels between participants with varying levels of education across age. Age ranges when differences are significant are highlighted with a red line on the x axis. The plots show that participants with high education have significantly higher exponents than those with lower education starting approximately from age 40 to 60 and until the end of the analyzed lifespan, except for the left temporal where significant differences went from 40 to 70 years.

**Table 4. ML and AIC scores (including differences between complex and simple models), along with effective degrees of freedom (Edf) and p-values assessing the significance of differences between the simple and complex offset models. Percentages of explained deviance are also reported. Model selection is only based on AIC differences greater than −2 (marked with *).**

| ROI | Explained Deviance | | ML scores | | | AIC | | | Edf | | | |
|---|---|---|---|---|---|---|---|---|---|---|---|---|
| | complex | simple | complex | simple | ML score difference (c-s) | complex | simple | AIC difference (c-s) | complex | simple | df | p value |
| left cingulate | 3.444 | 2.326 | 1429.412 | 1434.15 | 4.739 | 2859.866 | 2865.004 | −5.138* | 9 | 6 | 3 | 0.024 |
| right cingulate | 4.027 | 3.135 | 1426.044 | 1429.797 | 3.752 | 2854.716 | 2856.58 | −1.864 | 9 | 6 | 3 | 0.057 |
| left HPC | 8.354 | 8.154 | 1468.387 | 1472.689 | 4.302 | 2937.076 | 2940.167 | −3.091* | 9 | 6 | 3 | 0.035 |
| right HPC | 8.115 | 7.452 | 1456.577 | 1460.264 | 3.687 | 2914.148 | 2917.234 | −3.086* | 9 | 6 | 3 | 0.061 |
| left occipital | 8.012 | 8.037 | 1410.511 | 1411.781 | 1.271 | 2820.089 | 2818.237 | 1.852 | 9 | 6 | 3 | 0.468 |
| right occipital | 8.696 | 8.589 | 1417.529 | 1419.358 | 1.829 | 2836.67 | 2836.099 | 0.571 | 9 | 6 | 3 | 0.301 |
| left parietal | 9.713 | 9.423 | 1370.836 | 1372.999 | 2.163 | 2739.767 | 2740.018 | −0.251 | 9 | 6 | 3 | 0.228 |
| right parietal | 7.504 | 7.397 | 1409.345 | 1412.064 | 2.719 | 2820.273 | 2820.531 | −0.258 | 9 | 6 | 3 | 0.142 |
| left temporal | 11.133 | 11.021 | 1443.12 | 1444.784 | 1.664 | 2884.147 | 2883.116 | 1.031 | 9 | 6 | 3 | 0.344 |
| right temporal | 12.076 | 11.894 | 1440.388 | 1441.863 | 1.475 | 2876.361 | 2875.464 | 0.897 | 9 | 6 | 3 | 0.399 |

Age and education were observed to interact in modulating offset levels in a similar way across regions (Fig 4): younger age and higher education were associated with increased offset levels, mirroring trends observed for the exponent in the left cingulate.

Direct comparisons between individuals with higher and lower education levels revealed that older individuals with higher education exhibited significantly higher offset levels compared to their peers with lower education levels, an effect spanning from after 60 until 91 years of age; in addition, in the left cingulate, this relationship is reversed for young participants: in the age range of 18 to approximately 25 years, those with lower education have higher offsets than their same-age counterparts.

In summary, middle- and older-aged participants were found to have lower offsets in the left cingulate and bilateral HPC; similarly to what we found for the exponent, older participants with higher education reported significantly higher offsets compared to their peers with lower education levels.

**3.1.3. Aim 1 results summary.** Analyses aiming at exploring relationships between EEG components, age, and education revealed that in the left cingulate, bilateral HPC, left parietal and left temporal regions, older adults exhibited lower exponents (flatter slopes), but higher education mitigated the age-related decrease in exponent values. A similar pattern was observed for the offset, but only in the left cingulate and bilateral HPC.

## 3.2. Aim 2: aperiodic components, age, and education as predictors of MMSE scores

**3.2.1. Baseline MMSE model.** The baseline MMSE model revealed a significant interaction between age and education in influencing MMSE scores. As expected, younger participants and older participants with higher education had higher MMSE scores compared to older participants with lower education (yellow and green shades as opposed to darker blue shades in Fig 6a; see Fig 6b for direct comparisons between participants with higher and lower education). The difference between participants with higher and lower education was significant across the entire age range of interest (18–91 years old; Fig 6c).

**3.2.2. MMSE and exponent.** When examining the relationship between MMSE scores and exponent, a more complex model incorporating the interaction term between age, education, and exponent was preferred over the baseline MMSE model in all regions (Table 5), where the exponent influences the known interaction between age, education, and MMSE scores: for lower exponents (Fig 6a,d), the interaction effect mirrors that of the baseline

# Offset complex models: Age*Education interaction

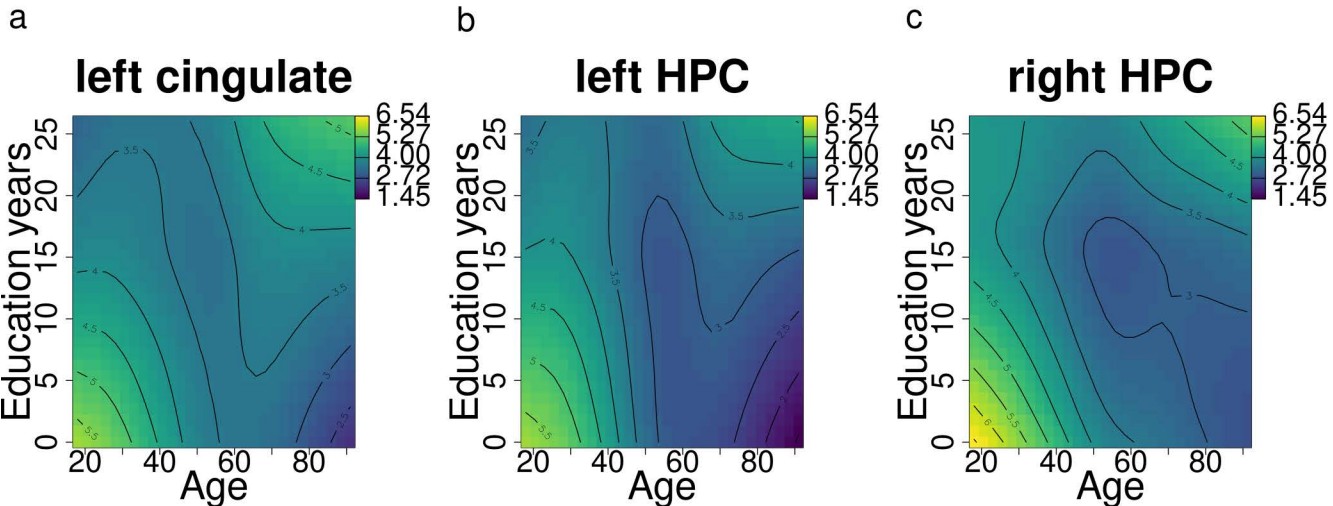

**Fig 4. Offset changes according to age and education (results of the complex models' interactions) for the left cingulate and bilateral HPC.** Darker blue shades indicate lower offsets, while lighter green and yellow shades indicate higher offsets.

MMSE model (Fig 6a), where younger age and older age with higher education are associated with higher MMSE scores. Moving towards median exponent values (Fig 7b,e), higher MMSE scores are observed even among older individuals with lower education; however, at higher exponent levels (Fig 7c,f), this association appears to reverse (especially for the left temporal): whereas older participants with higher education had higher MMSE scores at lower and median exponent levels, older individuals with higher exponents and higher education show worse MMSE scores compared to those with lower education. Since the effects are relatively consistent across all regions, only the figures for the left and right temporal areas are presented here, while figures for the remaining regions can be found in the Supporting Information.

When comparing individuals with high and low exponents across different education levels, this reversed association becomes clearer (Fig 8). Among participants with lower education, those with higher exponents have higher MMSE scores than those with lower exponents, particularly at middle and older ages (approximately >40; Fig 8a,c). In contrast, among participants with higher education, this pattern is reversed: participants with higher exponents show higher MMSE scores at younger ages (approximately <60), but lower MMSE scores at older ages (Fig 8b,d). These differences are significant across the entire age range of interest (18–91 years; Fig 8e-h). Another important observation is the greater variability in MMSE scores among participants with lower education compared to those with higher education. In contrast, in their counterparts with higher education, MMSE scores are clustered near the upper limit [30] until approximately age 60, after which scores of participants with higher exponents begin to decline.

In summary, in older adults with lower education, there is a possible positive relationship between exponent and MMSE scores (lower exponents correspond to lower scores), while those with higher education show the opposite trend (lower exponents correspond to higher MMSE scores).

**3.2.3. MMSE and offset.** When examining the relationship between MMSE scores and offset, the pattern of results closely mirrors the findings on MMSE and exponent. More complex models that included the interaction between age, education, and exponent were preferred over the baseline MMSE model in all regions (Table 6).

## Offset by education differences

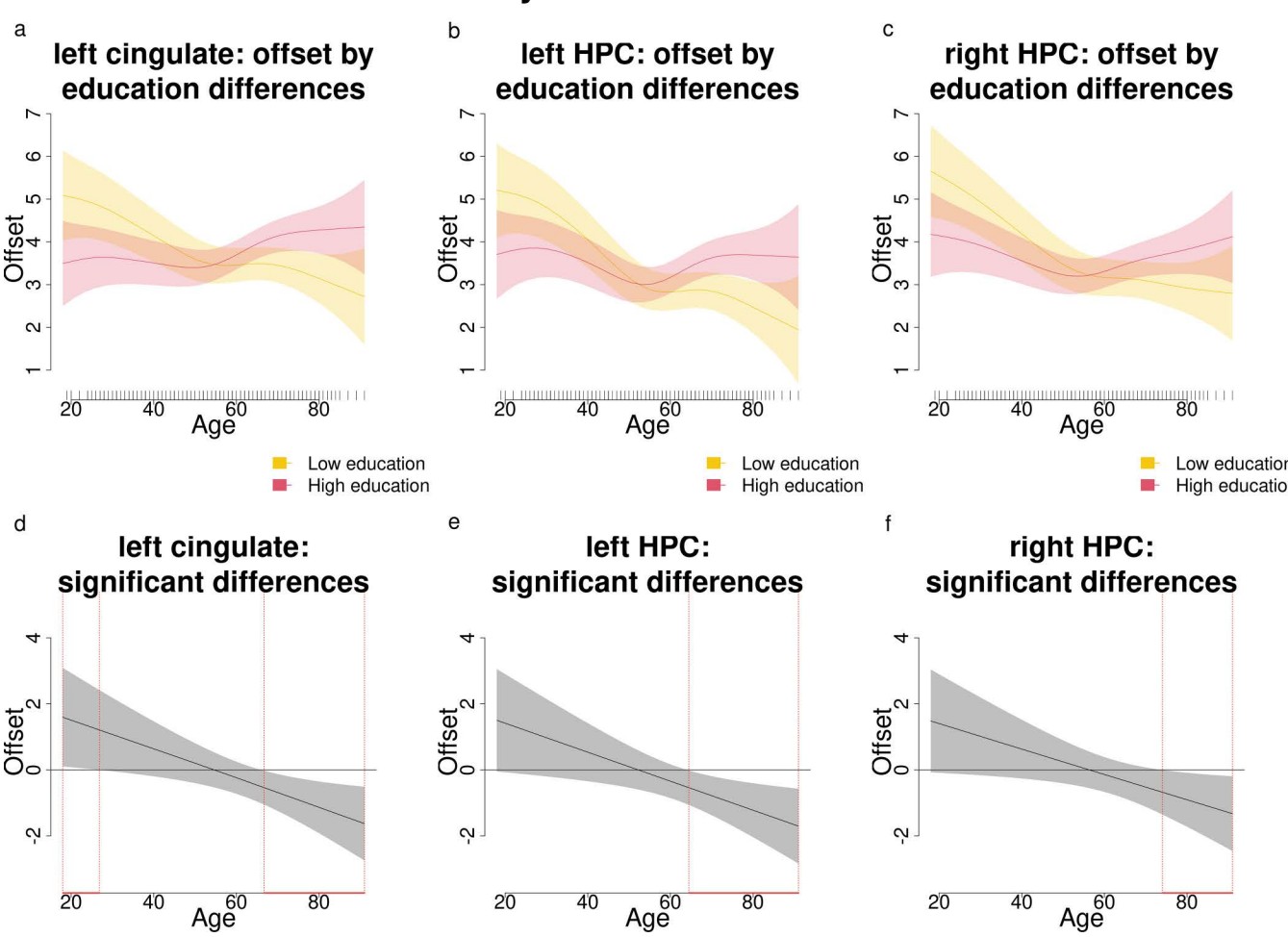

**Fig 5. Comparisons and significant differences between offset levels of participants with high and low education, in relation with age for the left cingulate and bilateral HPC.** Panels a-c show how age shapes the offset for different levels of education (high and low). Panels d-f show significant differences in offset between participants with varying levels of education across age. Age ranges when differences are significant are highlighted with a red line on the x axis. Panels b and c show that in the bilateral HPC, participants with high education have significantly higher offsets than those with lower education starting approximately from after age 60. In addition, panel a shows that in the age range from 18 to approximately 25, participants with lower education have significantly higher offsets than those with higher education.

As in the previous section, since the effects are relatively consistent across all regions, we report only the figures for the left and right temporal areas here, while figures for the remaining regions are available in the Supporting Information. Offset influences the established relationship between age, education, and MMSE scores in a manner similar to its effect on the relationship between MMSE and exponent. Specifically, at lower offset values (Fig 9a,d), the interaction effect resembles that of the baseline MMSE model (Fig 6a). As offset values increase toward the median (Fig 9b,e), higher MMSE scores are observed even among older individuals with lower education. However, at higher offset levels (Fig 9c,f), this association reverses (again, as for the exponent, especially for the left temporal): older individuals with both higher offsets and higher education exhibit lower MMSE scores compared to those with lower education.

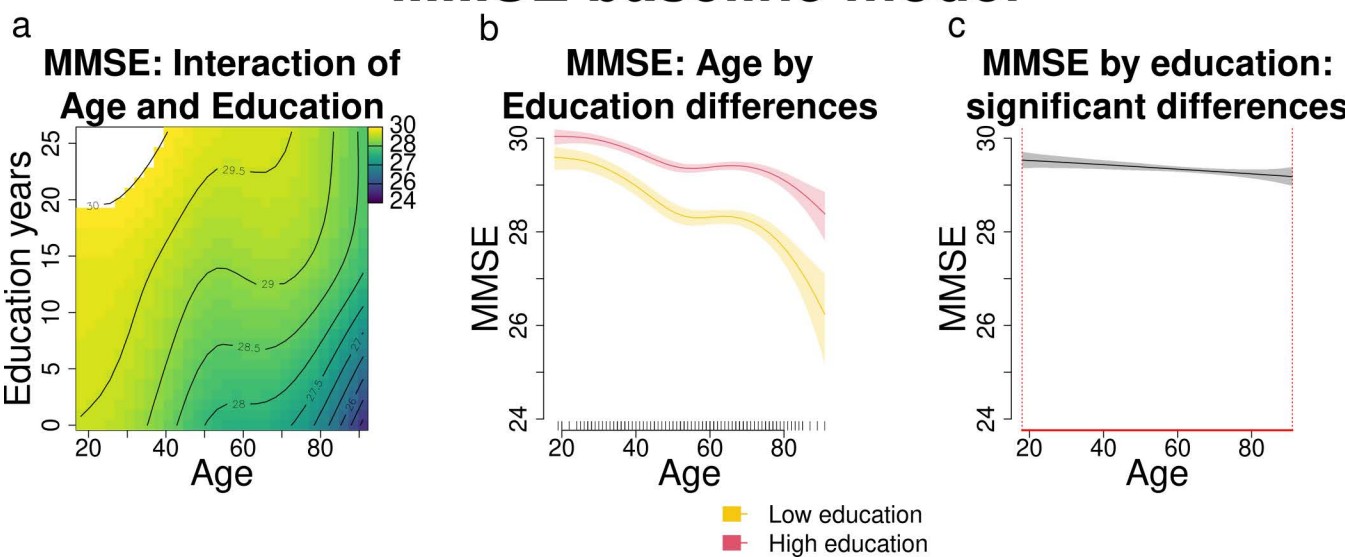

**Fig 6. a)** MMSE score changes according to age and education (results of the models' interaction). Darker blue shades indicate lower MMSE scores, while lighter green and yellow shades indicate higher scores. The white area indicates model estimates exceeding the upper MMSE score limit of 30. **b)** Comparison between MMSE scores of participants with high and low education, in relation with age: varying levels of education shape MMSE performance differently across age. **c)** Significant differences between MMSE scores of participants with high and low education, in relation with age. Age ranges when differences are significant are highlighted with a red line on the x axis.

When comparing participants with high and low offsets across different education levels (Fig 10), we observe distinct trends. Among participants with lower education, those with higher offsets tend to show better MMSE scores beginning around age 40 (Fig 10a,c). However, this pattern reverses in participants with higher education, and it is more evident in the left temporal area: individuals with higher offsets display higher MMSE scores until approximately age 60, after which their scores decline (Fig 10b,d). These differences are statistically significant across the entire age range of interest (18–91 years; Fig 10e-h) and closely parallel the relationship between MMSE and exponent across varying levels of age and education.

**3.2.4. Aim 2 results summary.** The results for the aperiodic exponent and offset in relation to MMSE scores were remarkably similar: in both cases, more complex models that included interaction terms between the aperiodic component, age, and education were preferred over the baseline MMSE model in all the regions under investigation. Additionally, the pattern of results remained consistent across both ROIs and aperiodic components.

More in detail, both the exponent and offset significantly predicted cognitive performance across age and education levels. In older adults with lower education, a positive relationship between the aperiodic components and MMSE scores was observed, with lower exponents and offsets corresponding to lower MMSE scores. In contrast, among those with higher education, a reverse trend was found, where lower exponents and offsets were associated with higher MMSE scores.

Given the remarkable similarity between exponent and offset results, we inspected the correlations between the two aperiodic components separately for each ROI through Spearman's correlation coefficient. Results are reported in Supplementary Table 6 and show that all correlations are significant and very high, ranging from 0.937 to 0.97. This finding demonstrates that exponent and offset share substantial variance and suggest that the present results are probably driven by such correlation.

**Table 5. ML and AIC scores (including differences between complex and simple models), along with effective degrees of freedom (Edf) and p-values assessing the significance of differences between the baseline MMSE model and complex models incorporating the interaction with the exponent. Percentages of explained deviance are also reported. Model selection is only based on AIC differences greater than −2 (marked with *).**

| | Explained Deviance | | ML scores | | | AIC | | | Edf | | | |
|---|---|---|---|---|---|---|---|---|---|---|---|---|
| ROI | complex | simple | complex | simple | ML score difference (c-s) | complex | simple | AIC difference (c-s) | complex | simple | df | p value |
| left cingulate | 21.356 | 18.957 | 562.278 | 614.551 | 52.274 | 1117.609 | 1215.951 | −98.342* | 18 | 9 | 9 | < 0.001 |
| right cingulate | 21.226 | 18.957 | 565.059 | 614.551 | 49.492 | 1124.11 | 1215.951 | −91.841* | 18 | 9 | 9 | < 0.001 |
| left HPC | 20.942 | 18.957 | 564.643 | 614.551 | 49.908 | 1120.28 | 1215.951 | −95.671* | 18 | 9 | 9 | < 0.001 |
| right HPC | 20.428 | 18.957 | 569.548 | 614.551 | 45.004 | 1128.117 | 1215.951 | −87.834* | 18 | 9 | 9 | < 0.001 |
| left occipital | 20.733 | 18.957 | 566.035 | 614.551 | 48.516 | 1125.02 | 1215.951 | −90.931* | 18 | 9 | 9 | < 0.001 |
| right occipital | 20.526 | 18.957 | 566.82 | 614.551 | 47.731 | 1125.37 | 1215.951 | −90.581* | 18 | 9 | 9 | < 0.001 |
| left parietal | 20.244 | 18.957 | 565.683 | 614.551 | 48.869 | 1125.18 | 1215.951 | −90.771* | 18 | 9 | 9 | < 0.001 |
| right parietal | 19.978 | 18.957 | 567.752 | 614.551 | 46.8 | 1128.74 | 1215.951 | −87.211* | 18 | 9 | 9 | < 0.001 |
| left temporal | 21.235 | 18.957 | 565.859 | 614.551 | 48.693 | 1123.846 | 1215.951 | −92.105* | 18 | 9 | 9 | < 0.001 |
| right temporal | 20.187 | 18.957 | 570.412 | 614.551 | 44.14 | 1131.488 | 1215.951 | −84.463 | 18 | 9 | 9 | < 0.001 |

## 4. Discussion

The present study addressed two main objectives: [1] to examine the relationships between EEG aperiodic components, age, and education, aiming to confirm known age-related changes while exploring the less-studied effects of education on aperiodic components; and [2] to investigate whether age, education, and EEG aperiodic components could predict cognitive performance, as measured by MMSE scores. In line with recent work, we interpret aperiodic parameters as sensitive but non-specific markers of circuit state rather than direct assays of excitation–inhibition (E:I), and we emphasize the importance of considering behavioral state and oscillatory markers alongside exponent and offset [11, 12].

Our findings align with the existing literature on age-related reductions in the exponent and offset, and on the association between reductions in the aperiodic components and worsening cognition [4, 9, 10, 13, 14, 21, 22, 24]. Results also revealed that age and education interacted in predicting the exponent in the left cingulate, bilateral hippocampus, left parietal and left temporal regions; age and education also predicted the offset in the left cingulate and bilateral hippocampus, extending the findings of previous studies [13, 14, 17, 27]. Notably, we found that individuals with higher educational attainment exhibited higher exponents and offsets than their less-educated peers starting approximately in the middle age (40 or later) and lasting until 91 years old, suggesting a possible protective effect of education against age-related declines in the exponent and offset. Given current debates, we note that age-related slope "flattening" and offset reduction can arise from multiple biophysical contributors (e.g., synaptic time scales, leak currents, nonlinear membrane dynamics) and should not be attributed exclusively to E:I alterations [11].

Our findings also add novel insights: both aperiodic components interacted with age and education in predicting MMSE scores in all the ROIs under investigation. Participants with lower education showed a positive relationship between aperiodic components and MMSE performance, with lower exponents and offsets predicting worse cognitive outcomes as age increased, starting around age 40. Conversely, participants with higher education displayed the opposite trend starting at around age 60, where lower aperiodic components were associated with better MMSE performance. This pattern of results should be considered with caution due to the low inter-individual variability in MMSE scores among highly educated younger and middle-aged participants, whose scores clustered near the upper limit [30]. In contrast, older participants and those with lower education exhibited greater variability in cognitive performance. This limited variability in MMSE scores among highly educated individuals may partially explain the observed differences in cognitive performance between participants with high and low education (see Figs 6b and 8a-d). Despite this limitation, our findings suggest

# MMSE and Exponent: Age*Education*Exponent interaction left and right temporal

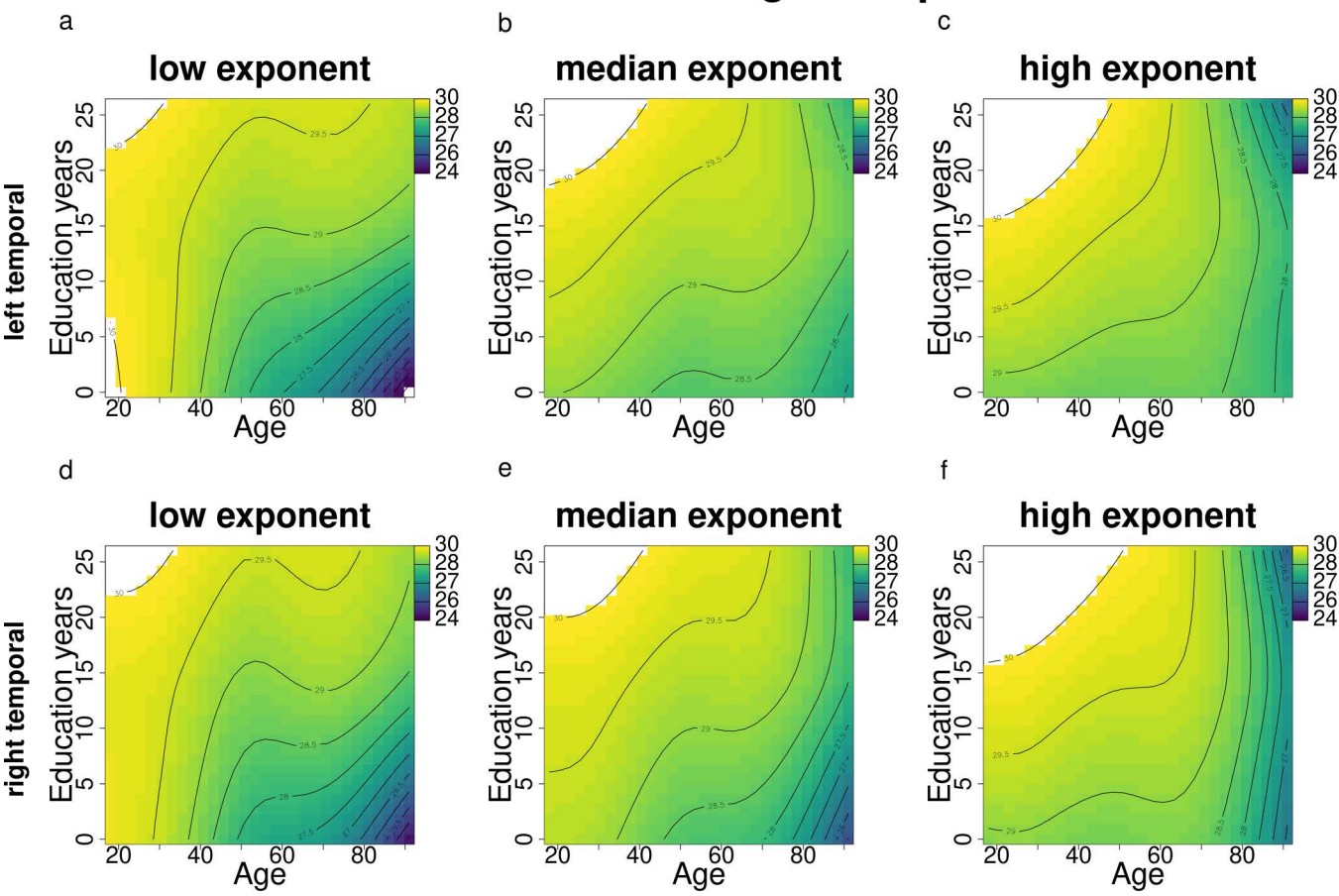

**Fig 7. MMSE score changes according to age and education across different exponent levels (results of the complex models' interactions) for the bilateral temporal regions.** Darker blue shades indicate lower MMSE scores, while lighter green and yellow shades indicate higher MMSE scores. The white areas indicate model estimates exceeding the upper MMSE score limit of 30.

that higher education likely modulates the association between aperiodic components and MMSE performance, and that these aspects could jointly act as protective factors against cognitive decline in older age. We frame these interactions cautiously, acknowledging that exponent/offset–cognition couplings need not reflect a single mechanism and may depend on state-related and task-related factors [12]. In interpreting our findings, it is important to clarify the relationship between the exponent and the offset. As expected from the spectral parameterization model, these two parameters were strongly correlated in our dataset (see Supplementary Table 6), and as stated also in the Introduction, this association reflects a well-described mathematical property of spectral rotation: when a change in the exponent induces a pivot around a non-zero frequency, the estimated offset shifts as well. Thus, the correlation between exponent and offset is not a competing finding with other interpretations stating that exponent and offset reflect separate neurophysiological underpinnings, but rather an inherent property of the PSD model [1, 7]. Because of this dependence, offset values cannot be interpreted as reflecting a distinct physiological process unless independent variation with respect to the exponent can be empirically demonstrated. In our data, the large overlap between exponent and offset-related findings in Aim 2, together with their

# MMSE by Exponent, Age and Education differences left and right temporal

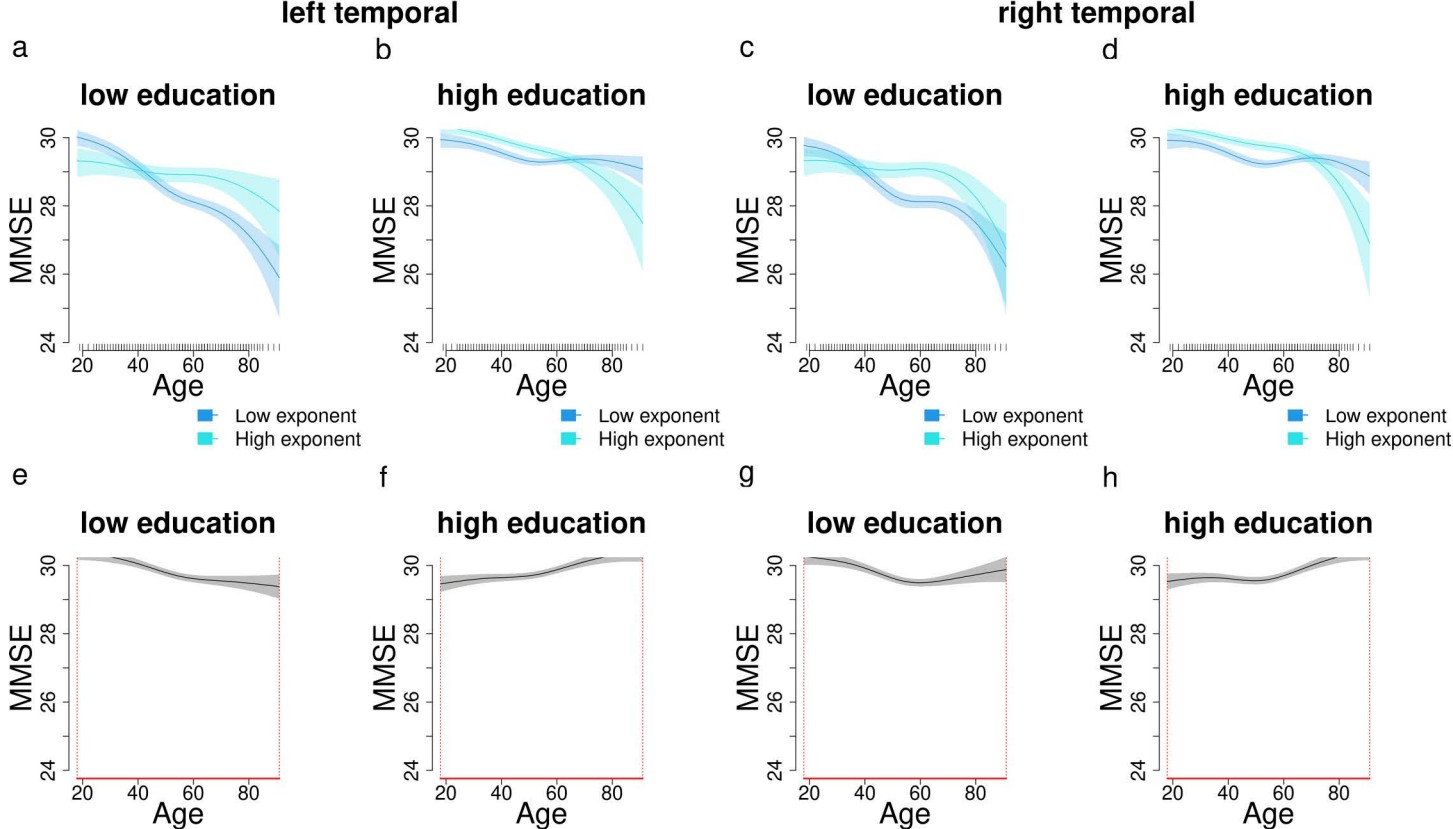

**Fig 8. Comparisons and significant differences between MMSE scores of participants with high and low exponents, in relation with age and different education levels for the left and right temporal regions.** Panels a-d show how different exponent levels shape MMSE scores across different levels of education (high and low) and increasing age. Panels e-h show significant differences in MMSE scores between participants with varying exponent and education levels across age. Age ranges when differences are significant are highlighted with a red line on the x axis. Panels a and c show that in the left and right temporal, participants with low education and higher exponents have significantly higher MMSE scores than those with lower exponents starting approximately from age 40. Panels b and d show that participants with higher education and higher exponents have worse MMSE scores than those with lower exponents. In panels e-h age ranges when differences are significant are highlighted with a red line on the x axis (MMSE score differences are significant across all the age window).

strong correlation, suggests that both parameters largely reflect the same underlying change in aperiodic activity. Even differences in Aim 1 findings, where the offset showed additional significant regions relative to the exponent, cannot be taken as evidence for separate neural mechanisms, as differences in the number of significant ROIs do not necessarily indicate meaningful dissociations between parameters. We therefore refrain from assigning distinct physiological interpretations to exponent and offset in the present study. For these reasons, we interpret our results in terms of aperiodic activity more generally, rather than ascribing independent neural processes to each parameter. This position is consistent with recent theoretical and modelling work showing that the exponent is not a universally reliable marker of excitation–inhibition balance [49], and with the fact that the biological interpretation of the offset remains less well established. Future studies explicitly designed to disentangle the relative contributions and potential independence of exponent and offset will be essential to advance our understanding of the physiological bases of aperiodic activity.

**Table 6.** ML and AIC scores (including differences between complex and simple models), along with effective degrees of freedom (Edf) and p-values assessing the significance of differences between the baseline MMSE model and complex models incorporating the interaction with the offset. Percentages of explained deviance are also reported. Model selection is only based on AIC differences greater than −2 (marked with *).

| | Explained Deviance | | ML scores | | | AIC | | | Edf | | | |
|---|---|---|---|---|---|---|---|---|---|---|---|---|
| ROI | complex | simple | complex | simple | ML score difference (c-s) | complex | simple | AIC difference (c-s) | complex | simple | df | p value |
| left cingulate | 21.288 | 18.957 | 561.179 | 614.551 | 53.372 | 1116.937 | 1215.951 | −99.014* | 18 | 9 | 9 | < 0.001 |
| right cingulate | 21.657 | 18.957 | 562.646 | 614.551 | 51.906 | 1120.554 | 1215.951 | −95.397* | 18 | 9 | 9 | < 0.001 |
| left HPC | 21.547 | 18.957 | 559.909 | 614.551 | 54.643 | 1111.543 | 1215.951 | −104.408* | 18 | 9 | 9 | < 0.001 |
| right HPC | 19.732 | 18.957 | 568.883 | 614.551 | 45.669 | 1131.736 | 1215.951 | −84.215* | 18 | 9 | 9 | < 0.001 |
| left occipital | 20.537 | 18.957 | 565.607 | 614.551 | 48.944 | 1125.736 | 1215.951 | −90.215* | 18 | 9 | 9 | < 0.001 |
| right occipital | 20.104 | 18.957 | 567.202 | 614.551 | 47.35 | 1126.99 | 1215.951 | −88.961* | 18 | 9 | 9 | < 0.001 |
| left parietal | 21.383 | 18.957 | 563.351 | 614.551 | 51.2 | 1122.463 | 1215.951 | −93.488* | 18 | 9 | 9 | < 0.001 |
| right parietal | 20.014 | 18.957 | 567.092 | 614.551 | 47.459 | 1127.905 | 1215.951 | −88.046* | 18 | 9 | 9 | < 0.001 |
| left temporal | 21.56 | 18.957 | 563.769 | 614.551 | 50.783 | 1122.055 | 1215.951 | −93.896* | 18 | 9 | 9 | < 0.001 |
| right temporal | 20.877 | 18.957 | 569.588 | 614.551 | 44.964 | 1133.499 | 1215.951 | −82.452* | 18 | 9 | 9 | < 0.001 |

Previous studies investigating the relationship between education and aperiodic components [26, 31] found no significant effects, even when using large datasets, including the one reanalyzed here [34]. Those studies primarily used linear regression and cluster-based correlation methods, which may not have captured the non-linear relationships instead identified by our approach. Our results instead align with and extend the findings by Montemurro et al. [18], despite methodological differences. While they grouped participants by age and education and employed linear regression, we treated age and education as continuous variables and used a flexible statistical model. Consistent with their findings, we observed that older adults with lower education levels benefited cognitively from higher exponents and offsets, while higher-educated older adults showed the opposite pattern.This divergence is compatible with frameworks in which the functional impact of aperiodic activity depends on individual traits and compensatory dynamics, rather than a fixed monotonic mapping to E:I.

Strictly regarding the MMSE performance across age (baseline MMSE model results), our results reaffirm the well-established link between higher education and better cognitive performance (Fig 6). According to a threshold model of cognitive decline, individuals with higher levels of education exhibit better cognitive functioning compared to those with lower education throughout the entire age span and are able to compensate for age-related cognitive declines over a longer period (their cognitive performance drops later in life) [32]. The cognitive reserve theory extends this view, adding that highly educated individuals might exhibit slower rates of cognitive decline [33, 50]. Figs 6b and panels b and d from Figs 8 and 10 show that higher-educated participants' MMSE scores clustered near the upper limit [30], while those with lower education exhibited greater variability, supporting the observation that on average, individuals with higher education also have a better MMSE performance throughout the entire age span. From age 60 onwards instead, the decline in MMSE scores was steeper in participants with lower education, consistent with the cognitive reserve framework, suggesting a protective role of education (related to further life experiences, e.g., complexity of the occupational level) that may influence the capacity of brain structure and functions to cope with normal and pathological age-related changes [33, 50].

Adding the aperiodic components to the baseline MMSE model (Section 3.2.2. and 3.2.3.; MMSE and exponent, MMSE and offset) reversed the association between education and cognitive performance in older age (Figs 8 and 10, panels b and d). Among older adults with lower education, higher exponents and offsets predicted better cognitive performance and less pronounced cognitive decline. In contrast, higher-educated older adults exhibited the reverse pattern: higher aperiodic components predicted worse performance and steeper decline, as observed by Montemurro et al. [18] for the exponent. These findings also

# MMSE and Offset: Age*Education*Offset interaction left and right temporal

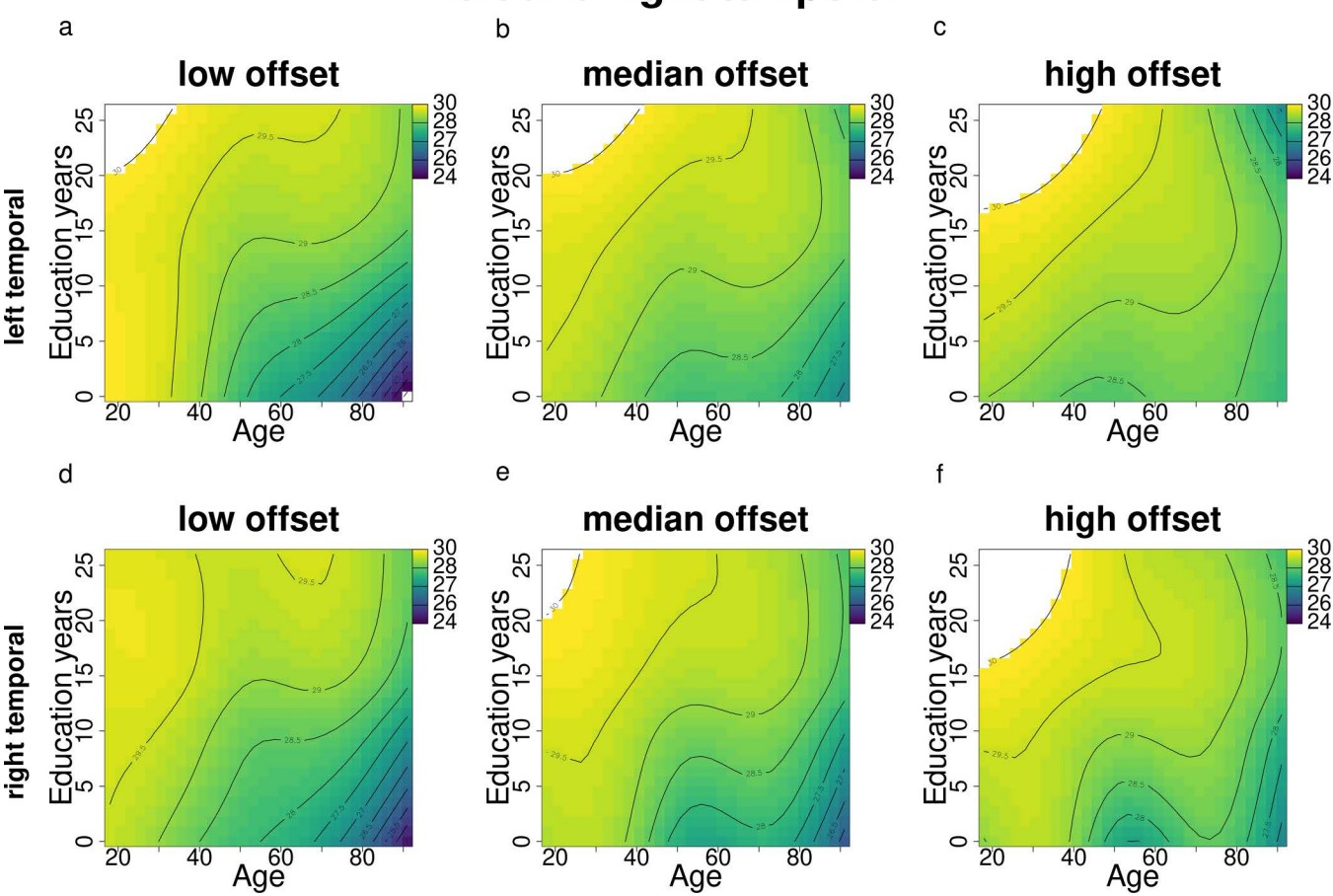

**Fig 9. MMSE score changes according to age and education across different offset levels (results of the complex models' interactions) for the bilateral temporal regions.** Darker blue shades indicate lower MMSE scores, while lighter green and yellow shades indicate higher MMSE scores. The white areas indicate model estimates exceeding the upper MMSE score limit of 30.

align with the neural noise hypothesis, which posits that reduced exponents reflect increased neural noise due to an increased E:I ratio during healthy aging [17]. Lower exponents are thus associated with flatter power spectra, decreased neural communication fidelity, and greater excitability (higher noise). These associations between aperiodic exponent and E:I ratio are supported by in-silico models [51] and studies manipulating the neural noise (measured with the aperiodic exponent) through the administration of pharmacological agents that either decrease (e.g., propofol) or increase excitation (ketamine) [51–53]. Age-related decreases in the offset observed here may be understood within accounts of increased aperiodic neural noise in aging [4, 6, 8, 17]. However, because exponent and offset share substantial variance due to rotation of the power spectrum, the offset cannot be interpreted as reflecting a distinct physiological mechanism unless independent variation is demonstrated. Our findings therefore align with previous reports of decreasing offset values across age [4, 8, 17, 18, 24], likely reflecting general changes in aperiodic activity rather than parameter-specific effects. Nonetheless, we explicitly acknowledge alternative viewpoints showing that the exponent may not track E:I under certain manipulations (e.g., subanesthetic ketamine, selective disinhibition), and thus we refrain from assigning a unique mechanistic interpretation to exponent changes in our data [12].

# MMSE by Offset, Age and Education differences left and right temporal

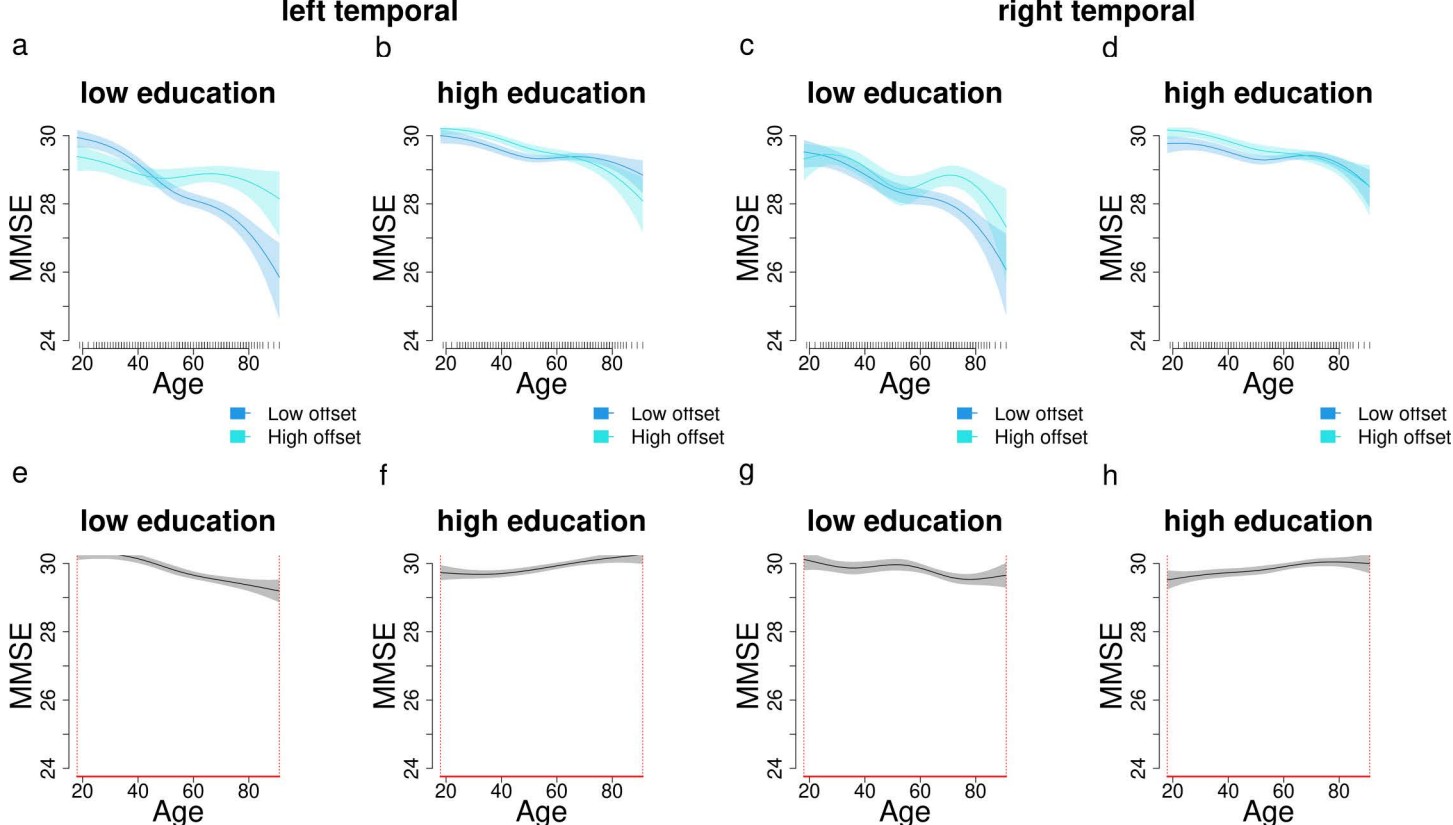

**Fig 10. Comparisons and significant differences between MMSE scores of participants with high and low offsets, in relation with age and different education levels for the left and right temporal regions.** Panels a-d show how different offset levels shape MMSE scores across different levels of education (high and low) and increasing age. Panels e-h show significant differences in MMSE scores between participants with varying offset and education levels across age. Age ranges when differences are significant are highlighted with a red line on the x axis. Panels a and c show that in the left and right cingulate, participants with low education and higher offsets have significantly higher MMSE scores than those with lower offsets starting approximately from age 40. Panels b and d show that participants with higher education and higher exponents have worse MMSE scores than those with lower exponents. In panels e-h age ranges when differences are significant are highlighted with a red line on the x axis (MMSE score differences are significant across all the age window).

Interestingly, the stochastic resonance framework [54] suggests that there may be no single "optimal" level of neural noise for cognitive performance. Instead, the effects of noise on cognition may depend on individual factors and compensatory dynamics, so that subjective changes in the aperiodic components, and thus in noise levels and E:I balance, might perturbate the cognitive system's functioning in an age- and education-dependent way, according to the individual's unique characteristics [55]. Even if further research is needed to better disentangle the causes of offset changes (mere correlation with the exponent or changes in neural spiking [4, 5, 17]), the neural noise and stochastic resonance hypotheses may complement the cognitive reserve theory in explaining how the interaction between the aperiodic components and education influences cognitive performance, particularly in older adults. Our findings show that in older individuals with lower education, lower exponents and offsets are associated with poorer MMSE performance. This is consistent with previous research [14, 16, 17, 24, 52] and parallels findings in younger participants, where lower exponents are linked

to poorer performance on complex tasks [16, 56]. This pattern likely reflects the physiological decline associated with aging. In contrast, highly educated older adults maintain preserved cognitive performance despite lower exponent and offset values. This suggests that individuals with greater cognitive reserve are better equipped to compensate for both normal and pathological age-related changes at the cognitive and neural level [33, 50]. In this context, lower aperiodic components—interpreted as increased neural noise —may reflect compensatory neural processes that are accessible to individuals with higher cognitive reserve but not to those with lower cognitive reserve. In other words, greater neural noise in highly educated individuals may indicate adaptive mechanisms that help sustain cognitive function, supporting the idea that higher educational attainment serves as a protective factor against cognitive and neural decline in older age. Future work should test these interpretations with designs that more directly monitor and control behavioral state (e.g., vigilance measures) and that jointly model periodic and aperiodic activity, given their partial dependence.

These considerations underscore that the relationship between aperiodic components, age, and cognition is not straightforward, but is mediated by individual traits [55] and especially by education, an important factor in influencing the resilience of the cognitive system to age-related changes. This nonlinearity justifies our use of a statistical approach capable of capturing complex interactions, emphasizing the need for considering individual differences with appropriate analytical tools when studying continuous age-related brain and cognitive changes [45].

The dataset we used encompasses a broad age range (18–91 years), spanning the entirety of adulthood, which enabled us to reliably investigate the neurodevelopmental dynamics of aperiodic components and their influence on cognitive performance from early adulthood to late life. However, while the temporal patterns provide valuable insights, the spatial distribution of our findings merits further discussion, as there is an intrinsic ambiguity in the literature regarding the expected topographies of the associations between cognitive performance and aperiodic features [26]. Age and education interacted in predicting the aperiodic components in the left cingulate, bilateral hippocampus, left temporal and left parietal regions; on the other hand, MMSE scores were significantly associated with these factors across the cortex.

At the present stage, we can only offer a tentative interpretation of this pattern, because the exact neurobiological basis of aperiodic components still remains unclear [26, 28]: the majority of studies cited in the present work either focused on electrode clusters [15, 16, 28, 56] or averaged all scalp electrodes [4, 8, 26, 27, 31], while some focused a-priori on the occipital region [17, 18] and others (conducted using magnetoencephalography) found that cognitive performance was associated with aperiodic components in a distributed way across the brain [57], in line with our findings. In addition, the study providing the currently used dataset [34] investigated aperiodic components on source-reconstructed EEG data, but aggregated smaller brain areas into composite ROIs (Table 2). Such aggregation reduces the spatial resolution and therefore poses some limitations to the interpretations of our results; another limitation regarding the spatial distribution of the effects is that the currently used dataset did not provide any information about aperiodic exponent and offset within the frontal areas. However, the main theoretical explanations of the functional role of aperiodic components, like the neural noise hypothesis [17], constitute frameworks of general, widespread neural mechanisms that are likely not limited to single brain areas; and in the same vein, global cognitive ability measured with a screening tool like the MMSE cannot be circumscribed to specific cortical regions. Further research is therefore needed to clearly establish whether the broad regions in which we found a significant association between MMSE score, aperiodic components and demographic variables are truly more sensitive to age-related aperiodic changes: this caution is necessary in that aging is associated with a widespread increase of low frequencies power, that can result in exponent decrease across the whole brain [9], possibly leading also to a decrease in offset through spectral rotation. Another specific caveat concerns the significant results in the hippocampus. Because Hernandez et al. [34] reported effects in this region, we included the hippocampus in our analyses and likewise observed significant effects bilaterally for both Aim 1 and Aim 2. However, it is known that EEG source reconstruction algorithms may be inaccurate in estimating activation of mesial areas [58], so for this reason, we have reported these results for completeness, but we stress that they should be regarded as speculative and uncertain and should be further investigated with techniques with better spatial resolution such as intracranial recordings, that proved effective in

studying the aperiodic components in the hippocampus [59]. Specific limitations of the dataset include the lack of information regarding the timing or sequence of data collection, as well as of aperiodic goodness of fit measures such as the $R^2$ and the use of the MMSE to assess cognitive performance, particularly in younger participants and those with higher education. For these individuals, the MMSE is relatively easy to complete, often resulting in a ceiling effect, where scores cluster near the upper limit [30], thereby reducing inter-individual variability. However, the MMSE was chosen because it remains a widely used cognitive screening tool in both clinical and research settings. Its broad availability, ease of administration, and brevity make it a practical option for initial cognitive assessment, especially in contexts requiring quick and accessible evaluations [60]. Future research should investigate individual differences in cognition using more comprehensive measures that assess a broader range of cognitive domains. A final limitation that it is important to consider is the correlational nature of the study that prevents precise interpretations on the direction of the effect. Although we can hypothesize that education shapes the relationship between aperiodic components and cognition, we cannot exclude the role of other unknown variables that moderate this relationship. This is an intrinsic limitation of cross-sectional studies on aging, that it is worth to underline.

In light of these considerations, we can hypothesize that education might modulate the impact of aging on aperiodic components in a widespread way across the cortex through mechanisms of cognitive reserve and network neuroplasticity: higher education may potentially help buffering against cognitive decline and preserving neural efficiency, promoting enhanced neural communication despite aging and greater resilience in distributed brain networks. This hypothesis merits further investigation, as well as the neurobiological basis of aperiodic components in general and the interplay between aperiodic components and demographic variables in shaping cognitive performance in specific brain areas.

In conclusion, our findings suggest that the relationship between aperiodic components and cognitive aging is complex and influenced by educational attainment, which may act as a protective factor against age-related decline throughout the cortex. These results align with the neural noise and stochastic resonance hypotheses [14, 54], emphasizing the importance of accounting for individual differences [55] when studying age-related changes in EEG aperiodic components and cognition. Future research should further explore the exact neurobiological basis of aperiodic components and better investigate the influence of such components on a broader range of cognitive domains. Consistent with current debates, we recommend interpreting exponent/offset jointly, avoiding a one-to-one mapping to E:I, ensuring comparable behavioral state across groups, and applying physiologically justified detrending when quantifying oscillatory power [11, 12].

## Supporting information

**S1 File. The following supporting information can be downloaded at: https://osf.io/x8dnp/?view_only=0c2060b-3853540c9a9d1360d38f7d2af.** Table S1. Significance of the smooth terms (interactions and main effects of age and education) for the exponent models; Table S2. Significance of the smooth terms (interactions and main effects of age and education) for the offset models; Table S3. Significance of the smooth terms (interactions and main effects of age and education) for the baseline MMSE model; Table S4. Significance of the smooth terms (interactions and main effects of age, education, and exponent) for the MMSE and exponent models; Table S5. Significance of the smooth terms (interactions and main effects of age, education, and offset) for the MMSE and offset models; Table S6. Spearman's correlation coefficients between exponent and offset for each ROI; Figure S1. Distribution of education years by age. The distribution of education levels is uniform across age; Figure S2. Main effect of age on exponent for the simple models of right cingulate, left and right occipital, right parietal, and right temporal; Figure S3. Main effect of education on exponent for the simple models of right cingulate, bilateral occipital, right parietal and right temporal; Figure S4. Main effect of age on offset for the simple models of right cingulate, bilateral occipital, bilateral parietal and bilateral temporal; Figure S5. Main effect of education on offset for the simple models of right cingulate, bilateral occipital, bilateral parietal and bilateral temporal; Figure S6. MMSE score changes according to age and education across different exponent levels for the bilateral cingulate; Figure S7. MMSE score changes according to age and education across different exponent levels for the bilateral

hippocampus; Figure S8. MMSE score changes according to age and education across different exponent levels for the bilateral occipital regions; Figure S9. MMSE score changes according to age and education across different exponent levels for the bilateral parietal regions; Figure S10. Comparisons and significant differences between MMSE scores of participants with high and low exponents, in relation with age and different education levels for the bilateral cingulate; Figure S11. Comparisons and significant differences between MMSE scores of participants with high and low exponent, in relation with age and different education levels for the bilateral hippocampus; Figure S12. Comparisons and significant differences between MMSE scores of participants with high and low exponent, in relation with age and different education levels for the bilateral occipital region; Figure S13. Comparisons and significant differences between MMSE scores of participants with high and low exponent, in relation with age and different education levels for the bilateral parietal regions; Figure S14. MMSE score changes according to age and education across different offset levels for the bilateral cingulate; Figure S15. MMSE score changes according to age and education across different offset levels for the bilateral hippocampus; Figure S16. MMSE score changes according to age and education across different offset levels for the bilateral occipital region; Figure S17. MMSE score changes according to age and education across different offset levels for the bilateral parietal regions; Figure S18. Comparisons and significant differences between MMSE scores of participants with high and low offsets, in relation with age and different education levels for the bilateral cingulate; Figure S19. Comparisons and significant differences between MMSE scores of participants with high and low offsets, in relation with age and different education levels for the bilateral hippocampus; Figure S20. Comparisons and significant differences between MMSE scores of participants with high and low offsets, in relation with age and different education levels for the bilateral occipital region; Figure S21. Comparisons and significant differences between MMSE scores of participants with high and low offsets, in relation with age and different education levels for the bilateral parietal region.
(DOCX)

## Acknowledgments

We thank Hernandez et al. [34] for providing the publicly available dataset.

## Author contributions

**Conceptualization:** Sara Lago, Giorgio Arcara.

**Data curation:** Sara Lago.

**Formal analysis:** Giorgio Arcara.

**Methodology:** Sara Lago, Sara Zago, Giorgio Arcara.

**Supervision:** Sonia Montemurro, Rocco Salvatore Calabrò, Maria Grazia Maggio, Serena Dattola, Ilaria Casetta, Giorgio Arcara.

**Validation:** Sara Lago, Sara Zago, Sonia Montemurro, Rocco Salvatore Calabrò, Maria Grazia Maggio, Serena Dattola, Ilaria Casetta.

**Visualization:** Sara Lago.

**Writing – original draft:** Sara Lago, Sonia Montemurro.

**Writing – review & editing:** Sara Lago, Sara Zago, Sonia Montemurro, Rocco Salvatore Calabrò, Maria Grazia Maggio, Serena Dattola, Ilaria Casetta, Giorgio Arcara.

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
