## [Decision Letter · Decision Letter 0]

25 Jul 2025

PLOS ONE

Dear Dr. Arcara,

Thank you for submitting your manuscript to PLOS ONE. After careful consideration, we feel that it has merit but does not fully meet PLOS ONE’s publication criteria as it currently stands. Therefore, we invite you to submit a revised version of the manuscript that addresses the points raised during the review process.

I agree that your study addresses an important and timely question regarding the relationship between aperiodic EEG activity, ageing, and education. The study is well-motivated and methodologically ambitious, the use of open data and advanced modelling techniques is commendable.

However, both reviewers have raised significant concerns. The core neural measures (exponent and offset) are treated as independent despite their strong correlation, leading to potentially misleading interpretations. Source-level findings in the hippocampus are speculative given the limitations of EEG localisation, yet are not appropriately caveated. Finally, the statistical approach (e.g., use of negative binomial GAMMs) lacks justification, and essential demographic and procedural context is missing.

In light of these concerns, I am inviting you to revise your manuscript substantially and address the points raised in full. I believe that with careful attention to these issues, your work can make a valuable contribution to the literature.

We look forward to receiving your revised manuscript.

Kind regards,

Giulio Contemori, Ph.D.

Academic Editor

PLOS ONE

4. Please include captions for your Supporting Information files at the end of your manuscript, and update any in-text citations to match accordingly. Please see our Supporting Information guidelines for more information: http://journals.plos.org/plosone/s/supporting-information .

5. Please upload a copy of Supporting Information Figure/Table/etc. Figure 1-19, Table 1-5 which you refer to in your text on page 30-31.

Reviewers' comments:

Reviewer's Responses to Questions

**Comments to the Author**

1. Is the manuscript technically sound, and do the data support the conclusions?

Reviewer #1: Yes

Reviewer #2: Yes

2. Has the statistical analysis been performed appropriately and rigorously?

Reviewer #1: Yes

Reviewer #2: Yes

3. Have the authors made all data underlying the findings in their manuscript fully available?

Reviewer #1: Yes

Reviewer #2: Yes

4. Is the manuscript presented in an intelligible fashion and written in standard English?

Reviewer #1: Yes

Reviewer #2: Yes

Reviewer #1: In this manuscript, Lago et al investigate the relationship between aperiodic neural activity as measured from EEG, aging, education, and cognition as measured by the MMSE. They do so using an openly available dataset for which these data and measures are already available, and apply a new analysis to investigate the potential modulation by education of age-related changes in aperiodic activity, and then how this relates to MMSE scores. Overall, I think this is an interesting study that adds some useful information about age-related changes in aperiodic activity using an appropriate dataset for the stated goals. The approach and analyses seem generally appropriate for the questions at hand – however, I find there are some additional elements that need to be addressed in terms of some aspects of the methods reporting, data analyses, and interpretations in order for this study to be ready for publication.

Methods reporting: as this study uses measures derived from an open-source dataset and previous analysis of said dataset, I think certain additional details from the original study need to be re-reported in this manuscript so that the reader can understand the key measures without necessarily having to refer back to the original study. For example, as the aperiodic measures as so key to the analysis, it should be reported directly what frequency range was fit. Additionally, it would be useful to note the fooof settings that were used and if any goodness of fit measures were evaluated and if any model fits were excluded, etc, to communicate if these measures can be considered reliable. For the subject data, it would also be useful to briefly note how the education variable is encoded, and add a brief description of the MMSE.

In the definition of the models, a reference or brief description should be added for the nomenclature used in the model definitions explaining, for example, what terms such as ’s()’, ‘ti()’, and ‘bs-‘re’’ refer to.

The model evaluations use a threshold of -2 for the AIC values to include a model in the main manuscript. It is unclear where this threshold value comes from / how it was chosen – this should be briefly described and motivated.

The analyses and results separately fit models and report results for the exponent and offset, with the results often being very similar between the two parameters. As is briefly mentioned in the Discussion, the exponent and offset are highly correlated, as any non-zero rotation point change of the exponent induces a corresponding change in offset (https://fooof-tools.github.io/fooof/visualizers.html#spectral-rotation). Despite the brief note in the Discussion, the analysis and discussion does not adequately address this, as the results and interpretation are largely presented as if these two parameters can be interpreted separately, even though the pattern of results appear to be very likely to be driven by the correlation between the features rather than reflect two different findings. For example, the significant regions are often very similar between the two parameters, and it is noted in Aim 2 that the results are remarkably similar. Even where some analyses are not completely the same (for example, additional regions are significant in Aim 1 for offset as compared to the exponent), this by itself does not indicate a different pattern of findings here, as a difference in significance does not indicate a significant difference, and it may well be the case that it is still the correlated changes in the features that the model is picking up, even if there is a difference in passing the significance threshold. Overall, for the analysis of the offset, to present and interpret this as something different than the exponent, this needs further analysis work (for example, controlling for exponent changes in the statistical analyses) and/or much more caveating of the results and interpretations since as presented there is no clear motivation that the offset results are not simply driven by the exponent changes, and if so, they add little value.

It’s interesting that in Aim 1, the results are in the hippocampus. This raises several interesting points that perhaps warrant some additional consideration / discussion. Firstly, as briefly acknowledged in the discussion, source projection of scalp EEG data to sub-cortical and mesial areas may not be robust – while the authors acknowledge this, they do not motivate if they consider these results robust and interpretable (and why) and/or to what extent this limitation renders these results speculative and uncertain.

Additionally, this finding of a hippocampal focal point in Aim 1 (if it can be taken at face value) is interesting as there is little analysis of aperiodic activity in sub-cortical or medial temporal regions, and there may be some differences to the typical analysis of cortical and/or channel data, as this is not a direct anatomical comparison to most of the cited work on age-related changes in aperiodic parameters in EEG. More could perhaps be said on this point, potentially noting and including this preprint, which is one example of looking at aperiodic activity in these regions (https://www.biorxiv.org/content/10.1101/2024.10.03.616418v1.abstract), as well as looking for and discussion any other work that may be relevant.

There are some issues with the reference list - for example 5 & 7 are the same paper; 18 cites a preprint when the published version of the paper is available and can / should be cited instead; and 18 & 20 appear to separately cite the preprint and published version of what are actually the same paper. This is what I noticed and not necessarily an exhaustive list - the authors should do a full check for making sure the references are properly done.

Reviewer #2: This manuscript presents a timely investigation into the moderating role of education on the relationship between EEG aperiodic components and cognitive ageing. The study addresses an important gap in the literature and employs advanced statistical methods (GAMMs), which are highly suitable for addressing the research questions posed. Below, I provide a limited number of minor suggestions to strengthen the manuscript further.

1. The introduction describes the aperiodic exponent as reflecting excitation/inhibition (E:I) balance. However, this interpretation is under debate see Brake et al., 2024; Salvatore et al., 2024. It would strengthen both the introduction and the discussion to acknowledge alternative viewpoints

2. While the readers are referred to Hernandez et al. for more detailed participant information, given the present manuscript focuses heavily on education and cognition (as measured by the MMSE), more explicit detail on how these variables were collected/what was the criteria etc., is warranted. For example, explicitly mentioning the country as 13 years of education may map to different qualifications internationally.

3. Given that the age range of participants is broad, including a histogram or at least summary statistics for the actual age distribution would significantly improve the clarity and interpretability of the results.

4. Was educational attainment balanced across the age spectrum? for accurate interpretability of the results, authors should include data (e.g., table or figure) or summary statistics describing how education levels were distributed by age.

5. Can the authors clarify the sequence/timing of data collection? I.e., were all measures collected in the same session, or across different days? Was there any systematic lag? This is particularly important if cognitive scores and EEG were not obtained in close temporal proximity. While the readers are referred to Hernandez et al., this information is important for the present manuscript.

6. Again, while readers are referred to Hernandez et al., it may be beneficial to explicitly state the duration of EEG recordings and any participant instructions, as this information is important context for evaluating the EEG results.

7. Can the authors report overall fit quality (R², MAE) for the aperiodic modelling across each of the 10 ROIs? This will help assess the robustness of spectral decomposition.

8. The manuscript mentions that the original 82 brain regions (from the AAL atlas) were grouped into 10 “composite regions of interest (ROIs)” using mean averaging for analysis. However, it does not spell out the explicit criteria for this aggregation. I think it would be beneficial to specify whether this was based on anatomical, functional, literature-derived, or data-driven criteria.

9. The rationale for choosing negative-binomial GAMMs is unclear. The authors state this is to address non-normality in residuals, however, why was this chosen over other methods? From my understanding, negative binomials are typically used for count data and less commonly applied to continuous data. Was non-normality/overdispersion in residuals frequent? I think adding more detail as to why negative-binomial GAMMs were chosen specifically would be beneficial.

10. Authors should clarify how sex was coded in the statistical models.

11. Authors should state the rationale for the choice of smoothing parameters (default k) in the GAMMs.

Subject to the minor revisions/suggestions above, this is a significant and well-executed contribution to the literature on cognitive ageing, EEG biomarkers, and cognitive reserve. The suggested additions will improve clarity, interpretation, reproducibility, and the manuscript’s value to the field.

References:

Brake N, Duc F, Rokos A et al. A neurophysiological basis for aperiodic EEG and the background spectral trend. Nat Commun 2024;15:1514.

Salvatore SV, Lambert P, Benz A et al. Periodic and aperiodic changes to cortical EEG in response to pharmacological manipulation. J Neurophysiol 2024;131:529–40.

**Do you want your identity to be public for this peer review?** For information about this choice, including consent withdrawal, please see our Privacy Policy

Reviewer #1: No

Reviewer #2: No

---

## [Author Response · Author response to Decision Letter 1]

8 Sep 2025

Reviewer #1: In this manuscript, Lago et al investigate the relationship between aperiodic neural activity as measured from EEG, aging, education, and cognition as measured by the MMSE. They do so using an openly available dataset for which these data and measures are already available, and apply a new analysis to investigate the potential modulation by education of age-related changes in aperiodic activity, and then how this relates to MMSE scores. Overall, I think this is an interesting study that adds some useful information about age-related changes in aperiodic activity using an appropriate dataset for the stated goals. The approach and analyses seem generally appropriate for the questions at hand – however, I find there are some additional elements that need to be addressed in terms of some aspects of the methods reporting, data analyses, and interpretations in order for this study to be ready for publication.

Methods reporting: as this study uses measures derived from an open-source dataset and previous analysis of said dataset, I think certain additional details from the original study need to be re-reported in this manuscript so that the reader can understand the key measures without necessarily having to refer back to the original study. For example, as the aperiodic measures as so key to the analysis, it should be reported directly what frequency range was fit. Additionally, it would be useful to note the fooof settings that were used and if any goodness of fit measures were evaluated and if any model fits were excluded, etc, to communicate if these measures can be considered reliable. For the subject data, it would also be useful to briefly note how the education variable is encoded, and add a brief description of the MMSE.

Answer: Thank you for this consideration. We now included the required information in the main text (lines 152-162). Unfortunately, it is not stated in Hernandez et al. whether any goodness of fit measures were used in the aperiodic models’ fit (this limitation is now reported in the Methods section, lines 202-203 and in the Discussion, lines 746-747). However, the authors conducted additional analyses on signal quality in general: the Overall Data Quality (OQD) index was calculated in the source space, using the methodology proposed by Zhao et al. (1). Continuous signal was segmented into 1-second epochs, and each epoch was labeled as 1 for low-quality epochs or 0 for high-quality epochs. The OQD represents the percentage of EEG epochs with good quality, ranging from 0 for signals where all epochs were classified as low quality, to 100 for signals where all epochs were classified as high quality. A regression model was then built to predict signal quality based on the number of channels, which yielded non-significant results, suggesting that the number of channels did not impact the reported metrics. No other variables possibly influencing quality of the signal were assessed, and this is certainly one of the limitations of the dataset we used.

In the definition of the models, a reference or brief description should be added for the nomenclature used in the model definitions explaining, for example, what terms such as ’s()’, ‘ti()’, and ‘bs-‘re’’ refer to.

Answer: In GAMMs, the syntax ‘s()’ is used to set in the model the main effect of a variable, while ‘ti()’ is used for setting an interaction effect of two or more variables. This information is now reported in lines 247-248. In the original models, the syntax ‘s(ID, bs="re")’ was used to set the random effects for subjects, but these effects were removed following the considerations of Reviewer 2 on the models’ structure.

The model evaluations use a threshold of -2 for the AIC values to include a model in the main manuscript. It is unclear where this threshold value comes from / how it was chosen – this should be briefly described and motivated.

Answer: Thank you for this consideration. This is now better clarified in lines 265-271.

The analyses and results separately fit models and report results for the exponent and offset, with the results often being very similar between the two parameters. As is briefly mentioned in the Discussion, the exponent and offset are highly correlated, as any non-zero rotation point change of the exponent induces a corresponding change in offset (https://fooof-tools.github.io/fooof/visualizers.html#spectral-rotation). Despite the brief note in the Discussion, the analysis and discussion does not adequately address this, as the results and interpretation are largely presented as if these two parameters can be interpreted separately, even though the pattern of results appear to be very likely to be driven by the correlation between the features rather than reflect two different findings. For example, the significant regions are often very similar between the two parameters, and it is noted in Aim 2 that the results are remarkably similar. Even where some analyses are not completely the same (for example, additional regions are significant in Aim 1 for offset as compared to the exponent), this by itself does not indicate a different pattern of findings here, as a difference in significance does not indicate a significant difference, and it may well be the case that it is still the correlated changes in the features that the model is picking up, even if there is a difference in passing the significance threshold. Overall, for the analysis of the offset, to present and interpret this as something different than the exponent, this needs further analysis work (for example, controlling for exponent changes in the statistical analyses) and/or much more caveating of the results and interpretations since as presented there is no clear motivation that the offset results are not simply driven by the exponent changes, and if so, they add little value.

Answer: Thank you for raising this point. We added a disclaimer elaborating on the shared aspects of the offset and exponent in the Introduction (lines 53-56) and in the Discussion (lines 603-625).

It’s interesting that in Aim 1, the results are in the hippocampus. This raises several interesting points that perhaps warrant some additional consideration / discussion. Firstly, as briefly acknowledged in the discussion, source projection of scalp EEG data to sub-cortical and mesial areas may not be robust – while the authors acknowledge this, they do not motivate if they consider these results robust and interpretable (and why) and/or to what extent this limitation renders these results speculative and uncertain.

Additionally, this finding of a hippocampal focal point in Aim 1 (if it can be taken at face value) is interesting as there is little analysis of aperiodic activity in sub-cortical or medial temporal regions, and there may be some differences to the typical analysis of cortical and/or channel data, as this is not a direct anatomical comparison to most of the cited work on age-related changes in aperiodic parameters in EEG. More could perhaps be said on this point, potentially noting and including this preprint, which is one example of looking at aperiodic activity in these regions (https://www.biorxiv.org/content/10.1101/2024.10.03.616418v1.abstract), as well as looking for and discussion any other work that may be relevant.

Answer: Thank you for this consideration and the one above, which we treat as one single point. We now added some caveats on the hippocampus results in the discussion, lines 737-745.

There are some issues with the reference list - for example 5 & 7 are the same paper; 18 cites a preprint when the published version of the paper is available and can / should be cited instead; and 18 & 20 appear to separately cite the preprint and published version of what are actually the same paper. This is what I noticed and not necessarily an exhaustive list - the authors should do a full check for making sure the references are properly done.

Answer: Thank you for raising this point. We checked the reference list and now it should be correct.

Reviewer #2: This manuscript presents a timely investigation into the moderating role of education on the relationship between EEG aperiodic components and cognitive ageing. The study addresses an important gap in the literature and employs advanced statistical methods (GAMMs), which are highly suitable for addressing the research questions posed. Below, I provide a limited number of minor suggestions to strengthen the manuscript further.

1. The introduction describes the aperiodic exponent as reflecting excitation/inhibition (E:I) balance. However, this interpretation is under debate see Brake et al., 2024; Salvatore et al., 2024. It would strengthen both the introduction and the discussion to acknowledge alternative viewpoints

Answer: Thank you for raising this point. We added some considerations throughout the Introduction and Discussion detailing alternative views of the aperiodic components of the EEG spectrum, also following suggestions from Reviewer 1. This helped us put our work further into context and to broaden the interpretation of our results.

2. While the readers are referred to Hernandez et al. for more detailed participant information, given the present manuscript focuses heavily on education and cognition (as measured by the MMSE), more explicit detail on how these variables were collected/what was the criteria etc., is warranted. For example, explicitly mentioning the country as 13 years of education may map to different qualifications internationally.

Answer: Thank you for this consideration. More details on MMSE and education variables are now reported in lines 152-162.

3. Given that the age range of participants is broad, including a histogram or at least summary statistics for the actual age distribution would significantly improve the clarity and interpretability of the results.

Answer: Thank you for raising this point (and the one below). We added histograms for age and education in section 2.1.

4. Was educational attainment balanced across the age spectrum? for accurate interpretability of the results, authors should include data (e.g., table or figure) or summary statistics describing how education levels were distributed by age.

Answer: We clarified in the text that education levels were uniformly distributed across age (line 156-157) and added the relative plot in the Supplementary Information (Figure S1).

5. Can the authors clarify the sequence/timing of data collection? I.e., were all measures collected in the same session, or across different days? Was there any systematic lag? This is particularly important if cognitive scores and EEG were not obtained in close temporal proximity. While the readers are referred to Hernandez et al., this information is important for the present manuscript.

Answer: Unfortunately, in the original paper there was no information regarding the timing or sequence of data collection. This is certainly a limitation of the dataset.

6. Again, while readers are referred to Hernandez et al., it may be beneficial to explicitly state the duration of EEG recordings and any participant instructions, as this information is important context for evaluating the EEG results.

Answer: Thank you for this consideration. This information is now reported in lines 175-179.

7. Can the authors report overall fit quality (R², MAE) for the aperiodic modelling across each of the 10 ROIs? This will help assess the robustness of spectral decomposition.

Answer: Thank you for raising this point. A similar concern was also reported by Reviewer 1. It is not stated in Hernandez et al. whether any goodness of fit measures were used in the aperiodic models’ fit (this limitation is now reported in the Methods section, lines 202-203 and in the Discussion, lines 746-747). However, the authors conducted additional analyses on signal quality in general: the Overall Data Quality (OQD) index was calculated in the source space, using the methodology proposed by Zhao et al. (1). Continuous signal was segmented into 1-second epochs, and each epoch was labeled as 1 for low-quality epochs or 0 for high-quality epochs. The OQD represents the percentage of EEG epochs with good quality, ranging from 0 for signals where all epochs were classified as low quality, to 100 for signals where all epochs were classified as high quality. A regression model was then built to predict signal quality based on the number of channels, which yielded non-significant results, suggesting that the number of channels did not impact the reported metrics. No other variables possibly influencing quality of the signal were assessed, and this is certainly one of the limitations of the dataset we used.

8. The manuscript mentions that the original 82 brain regions (from the AAL atlas) were grouped into 10 “composite regions of interest (ROIs)” using mean averaging for analysis. However, it does not spell out the explicit criteria for this aggregation. I think it would be beneficial to specify whether this was based on anatomical, functional, literature-derived, or data-driven criteria.

Answer: Thank you for this consideration. The criteria for ROI aggregation is now clarified in lines 193-200.

9. The rationale for choosing negative-binomial GAMMs is unclear. The authors state this is to address non-normality in residuals, however, why was this chosen over other methods? From my understanding, negative binomials are typically used for count data and less commonly applied to continuous data. Was non-normality/overdispersion in residuals frequent? I think adding more detail as to why negative-binomial GAMMs were chosen specifically would be beneficial.

Answer: We thank the Reviewer for this consideration, which helped us to improve our models. Non-normality and overdispersion of residuals were indeed an issue, and choosing a negative-binomial family seemed to improve this aspect. However, following the reviewer’s comment and upon a better scrutiny of implication of our choice, we made some other attempts to fit the models with other distributions and with or without random effects. This strategy showed that fitting the models using a Gaussian family with identity link and without any random effect structure drastically improved the non-normality/overdispersion issues, instead of choosing the negative-binomial family and introducing random effects. Therefore, results of these improved models are now reported in the manuscript and these modeling choices are now better outlined in lines 241-245.

10. Authors should clarify how sex was coded in the statistical models.

Answer: Sex was coded as a factor and contrasts were set to 0=F (female) and 1=M (male). This is now clarified in line 246-247

11. Authors should state the rationale for the choice of smoothing parameters (default k) in the GAMMs.

Answer: Thank you for this consideration. For each model, we had 714 observations, 1 for each subject, and no random effects, so given the not-too-complex structure of the models we believe that setting k to the default could help avoiding overfitting. To check that such k setting was adequate following the Reviewer’s concern, we then re-fit the models doubling the value of k, and inspected the models’ residuals structure (using the gam.check() function) to see if there was any pattern in the residuals that could potentially be explained by increasing k as advised in https://stat.ethz.ch/R-manual/R-devel/library/mgcv/html/choose.k.html and in Wood (2). In no case increasing k provided benefits in removing any structure in the residuals, so we went on to use the default k (now reported in line 252-258).

Subject to the minor revisions/suggestions above, this is a significant and well-executed contribution to the literature on cognitive ageing, EEG biomarkers, and cognitive reserve. The suggested additions will improve clarity, interpretation, reproducibility, and the manuscript’s value to the field.

References:

Brake N, Duc F, Rokos A et al. A neurophysiological basis for aperiodic EEG and the background spectral trend. Nat Commun 2024;15:1514.

Salvatore SV, Lambert P, Benz A et al. Periodic and aperiodic changes to cortical EEG in response to pharmacological manipulation. J Neurophysiol 2024;131:529–40.

Response refere

---

## [Decision Letter · Decision Letter 1]

29 Sep 2025

Dear Dr. Arcara,

Thank you for submitting your manuscript to PLOS ONE. After careful consideration, we feel that it has merit but does not fully meet PLOS ONE’s publication criteria as it currently stands. Therefore, we invite you to submit a revised version of the manuscript that addresses the points raised during the review process.

The authors have satisfactorily addressed the major concerns and made revisions to the manuscript that incorporate several of the original comments. However, a few minor points persist, most notably concerning the interpretation of the offset and exponent as independent parameters.

We look forward to receiving your revised manuscript.

Kind regards,

Giulio Contemori, Ph.D.

Academic Editor

PLOS ONE

Journal Requirements:

Reviewers' comments:

Reviewer's Responses to Questions

**Comments to the Author**

Reviewer #1: (No Response)

Reviewer #2: All comments have been addressed

2. Is the manuscript technically sound, and do the data support the conclusions?

Reviewer #1: Yes

Reviewer #2: Yes

3. Has the statistical analysis been performed appropriately and rigorously?

Reviewer #1: Yes

Reviewer #2: Yes

4. Have the authors made all data underlying the findings in their manuscript fully available?

Reviewer #1: Yes

Reviewer #2: Yes

5. Is the manuscript presented in an intelligible fashion and written in standard English?

Reviewer #1: Yes

Reviewer #2: Yes

Reviewer #1: I have read the response to the reviewers and the update to the manuscript. Broadly, the authors have responded to main points and made edits to the manuscript that address several of the original comments. However, in doing so I find there to be some issues with how updates to the manuscript are phrased and presented. For example, the authors seem to mis-represent some points about the relationship between the exponent and the offset, as if there are two competing viewpoints about the offset (that it is correlated with the exponent vs it being a separate phenomenon). These are not opposing views - the correlation of the offset and exponent is a finding, not an interpretation, based on the offset being impacted by any non-zero frequency rotation point change in exponent. It is also true that the offset can change and be interpreted separately from the exponent, if one analyzes for this situation. Without demonstrating this, however, the offset cannot be interpreted separately from the offset. Related to this, some of the phrasing is poor. For example, it is not clear what “exponent and offset are highly correlated due to their shared influence on the power spectrum” is actually supposed to mean. The offset and the exponent do not have shared influence on the PSD, they are measures of the PSD. These descriptions do not seem to reflect a clear perspective and interpretation of the phenomena under study. It is also not clear why authors do not report the any correlation measures between offset and exponent in this dataset to speak to this topic in a more empirical sense. Some of the other edits are also not particularly precise in relation to the literature - for example, I am not clear why the Salvatore paper, which is largely about pharmacological manipulations, is cited in the introduction as if the main comparison relates to sleep vs wakefulness. Overall, I think the authors need to ensure they revisit the related literature and their framing of it to make sure their descriptions of features of aperiodic activity are clear and well-described.

Reviewer #2: I have carefully reviewed the authors’ revisions and responses to my comments. Overall, I am satisfied with the changes, which have substantially improved the clarity and interpretability of the manuscript. I have only a few remaining minor recommendations: (i) add a reference to support the MMSE statement on line 162 (“generally, a score of 24 or higher indicates normal cognitive functioning”), (ii) label the x-axis in the education histograms with units (“years”), including in the supplementary figures, and (iii) explicitly acknowledge the lack of information on the timing/sequence of data collection as a limitation. With these minor revisions addressed, I am happy to recommend the manuscript for publication.

**Do you want your identity to be public for this peer review?** For information about this choice, including consent withdrawal, please see our Privacy Policy

Reviewer #1: No

Reviewer #2: No

---

## [Author Response · Author response to Decision Letter 2]

15 Dec 2025

Reviewer #1: I have read the response to the reviewers and the update to the manuscript. Broadly, the authors have responded to main points and made edits to the manuscript that address several of the original comments. However, in doing so I find there to be some issues with how updates to the manuscript are phrased and presented. For example, the authors seem to mis-represent some points about the relationship between the exponent and the offset, as if there are two competing viewpoints about the offset (that it is correlated with the exponent vs it being a separate phenomenon). These are not opposing views - the correlation of the offset and exponent is a finding, not an interpretation, based on the offset being impacted by any non-zero frequency rotation point change in exponent. It is also true that the offset can change and be interpreted separately from the exponent, if one analyzes for this situation. Without demonstrating this, however, the offset cannot be interpreted separately from the offset. Related to this, some of the phrasing is poor. For example, it is not clear what “exponent and offset are highly correlated due to their shared influence on the power spectrum” is actually supposed to mean.

The offset and the exponent do not have shared influence on the PSD, they are measures of the PSD. These descriptions do not seem to reflect a clear perspective and interpretation of the phenomena under study. It is also not clear why authors do not report the any correlation measures between offset and exponent in this dataset to speak to this topic in a more empirical sense. Some of the other edits are also not particularly precise in relation to the literature - for example, I am not clear why the Salvatore paper, which is largely about pharmacological manipulations, is cited in the introduction as if the main comparison relates to sleep vs wakefulness. Overall, I think the authors need to ensure they revisit the related literature and their framing of it to make sure their descriptions of features of aperiodic activity are clear and well-described.

Answer: Thank you for raising these points, that helped us improve our phrasing and clarify our point of view. We will answer here point by point.

The authors seem to mis-represent some points about the relationship between the exponent and the offset, as if there are two competing viewpoints about the offset (that it is correlated with the exponent vs it being a separate phenomenon) etc.

We chose to run the same analyses both exponent and offset a) because they refer to different measures of the PSD, and b) to be consistent with the literature that considers these two aperiodic components as different but related aspects of the background neural activity. For these reasons, throughout the paper we explicitly acknowledge that we are not trying to interpret the offset as separate from the exponent without sufficient support from our analyses and findings (e.g., lines 47-61 “... exponent-offset correlations are not competing with interpretations of these components as having separate neural generators, but rather expected properties of the model…”; lines 148-157: “we quantified the aperiodic exponent and offset separately, but we interpret them jointly, acknowledging their strong interdependence due to the mathematical property of power spectrum rotation… We therefore refrain from assigning parameter-specific physiological mechanisms and instead frame our results in terms of general aperiodic dynamics, consistent with multiple plausible biophysical interpretations…”; lines 624-648: “... offset values cannot be interpreted as reflecting a distinct physiological process unless independent variation with respect to the exponent can be empirically demonstrated…”; lines 690-696: “... because exponent and offset share substantial variance due to rotation of the power spectrum, the offset cannot be interpreted as reflecting a distinct physiological mechanism unless independent variation is demonstrated…”. However, we also discuss the possible neurophysiological meaning of exponent and offset in line with the existing literature that does not interpret these two aperiodic components as completely overlapping.

It is also not clear why authors do not report the any correlation measures between offset and exponent in this dataset to speak to this topic in a more empirical sense.

We computed correlation measures between exponent and offset (now reported in the Supplementary Materials) supporting their consideration that our findings are driven by the amount of shared variance between the two aperiodic components (lines 579-584; line 626). This additional finding does not alter the paper’s conclusions but further supports the idea that exponent and offset are closely related and are not easily distinguishable.

Some of the other edits are also not particularly precise in relation to the literature - for example, I am not clear why the Salvatore paper, which is largely about pharmacological manipulations, is cited in the introduction as if the main comparison relates to sleep vs wakefulness.

We rephrased the context in which the Salvatore et al. paper is cited (lines 79), so that now it is clearer that it refers to pharmacological manipulations.

Reviewer #2: I have carefully reviewed the authors’ revisions and responses to my comments. Overall, I am satisfied with the changes, which have substantially improved the clarity and interpretability of the manuscript. I have only a few remaining minor recommendations: (i) add a reference to support the MMSE statement on line 162 (“generally, a score of 24 or higher indicates normal cognitive functioning”), (ii) label the x-axis in the education histograms with units (“years”), including in the supplementary figures, and (iii) explicitly acknowledge the lack of information on the timing/sequence of data collection as a limitation. With these minor revisions addressed, I am happy to recommend the manuscript for publication.

Answer: All points have been addressed in the main text (point i, line 174; point iii, lines 190-191 and 771-772) and in the Supplementary Materials (point ii). Thank you for recommending the manuscript for publication.

---

## [Decision Letter · Decision Letter 2]

8 Feb 2026

Education Shapes the Link Between EEG Aperiodic Components and Cognitive Aging

PONE-D-25-31877R2

Dear Dr. Arcara,

We’re pleased to inform you that your manuscript has been judged scientifically suitable for publication and will be formally accepted for publication once it meets all outstanding technical requirements.

Kind regards,

Wang Zhan, Ph.D.

Academic Editor

PLOS One

Additional Editor Comments (optional):

Reviewers' comments:

Reviewer's Responses to Questions

**Comments to the Author**

Reviewer #1: All comments have been addressed

Reviewer #2: All comments have been addressed

2. Is the manuscript technically sound, and do the data support the conclusions?

Reviewer #1: Yes

Reviewer #2: Yes

3. Has the statistical analysis been performed appropriately and rigorously?

Reviewer #1: Yes

Reviewer #2: Yes

4. Have the authors made all data underlying the findings in their manuscript fully available?

Reviewer #1: Yes

Reviewer #2: Yes

5. Is the manuscript presented in an intelligible fashion and written in standard English?

Reviewer #1: Yes

Reviewer #2: Yes

Reviewer #1: I have read the author's responses to the reviewer comments and their updates to the paper, and find that they have addressed all of my outstanding comments. I have nothing further to add or suggest. Congratulations to the authors on contributing a useful and interesting paper to the literature!

Reviewer #2: I am satisfied that the authors have addressed my feedback and suggestions. I am happy to recommend this article for acceptance and publication.

**Do you want your identity to be public for this peer review?** For information about this choice, including consent withdrawal, please see our Privacy Policy

Reviewer #1: No

Reviewer #2: No

---

## [Editor Report · Acceptance letter]

PONE-D-25-31877R2

PLOS One

Dear Dr. Arcara,

I'm pleased to inform you that your manuscript has been deemed suitable for publication in PLOS One. Congratulations! Your manuscript is now being handed over to our production team.

Kind regards,

on behalf of

Dr. Wang Zhan

Academic Editor

PLOS One